# Optimized CRISPR tools and site-directed transgenesis towards gene drive development in *Culex quinquefasciatus* mosquitoes

Xuechun Feng [1], Víctor López Del Amo [1], Enzo Mameli [2,3], Megan Lee[1], Alena L. Bishop [1], Norbert Perrimon [2,4] & Valentino M. Gantz [1✉]

*Culex* mosquitoes are a global vector for multiple human and animal diseases, including West Nile virus, lymphatic filariasis, and avian malaria, posing a constant threat to public health, livestock, companion animals, and endangered birds. While rising insecticide resistance has threatened the control of *Culex* mosquitoes, advances in CRISPR genome-editing tools have fostered the development of alternative genetic strategies such as gene drive systems to fight disease vectors. However, though gene-drive technology has quickly progressed in other mosquitoes, advances have been lacking in *Culex*. Here, we develop a *Culex*-specific Cas9/ gRNA expression toolkit and use site-directed homology-based transgenesis to generate and validate a *Culex quinquefasciatus* Cas9-expressing line. We show that gRNA scaffold variants improve transgenesis efficiency in both *Culex quinquefasciatus* and *Drosophila melanogaster* and boost gene-drive performance in the fruit fly. These findings support future technology development to control *Culex* mosquitoes and provide valuable insight for improving these tools in other species.

[1] Section of Cell and Developmental Biology, University of California San Diego, La Jolla, CA, USA. [2] Department of Genetics, Blavatnik Institute, Harvard Medical School, Boston, MA, USA. [3] Department of Microbiology, National Emerging Infectious Diseases Laboratories, Boston University, School of Medicine, Boston, MA, USA. [4] HHMI, Harvard Medical School, Boston, MA, USA. ✉email: vgantz@ucsd.edu

*C*ulex mosquitoes are widespread global vectors for several human and animal pathogens, including West Nile virus (WNV), Japanese encephalitis virus, the worm causing lymphatic filariasis, and the parasite causing avian malaria[1]. Several of these *Culex*-borne diseases, particularly West Nile and lymphatic filariasis, pose a significant risk to human health. WNV hospitalizations in the US alone impose an average of ~$56 M per year in health care costs[2], resulting in several annual deaths and thousands of diagnosed cases[3,4]. Lymphatic filariasis is a major public health burden in developing countries, where advanced stages of the disease can cause the chronic debilitating condition elephantiasis[5,6]. Within the genus, *Culex quinquefasciatus* has the greatest impact on human health due to its widespread distribution in urban and suburban areas and larval tolerance to polluted water reservoirs associated with human and livestock populations[7,8]. Its ability to hybridize with other species makes it adaptable to new environments[9], making this mosquito a challenging vector to control. In addition, *Culex quinquefasciatus* is the primary vector for avian malaria and avian pox, posing existential threats to island avifauna[10–12]. As an example, the invasive *Culex quinquefasciatus* in Hawai'i has contributed to the extinction of several Honeycreeper species and continues to threaten other susceptible species on the islands[13,14].

Current insecticide-based mosquito control strategies are beginning to fail due to the development of resistance in *Culex* populations[15,16]. Fortunately, the advent of CRISPR has allowed for the development of alternative genetic–engineering-based strategies that prevent disease transmission or suppress vector populations[17,18]. Although gene drives[19,20] and genetic sterile insect technology (gSIT)[21] are being successfully developed in *Anopheles*[20,22–25] and *Aedes*[26] mosquitoes, similar applications have lagged in *Culex*. Only recently have studies shown that CRISPR editing of the *Culex* genome is feasible using either embryo microinjection[27–29] or REMOT[30]. Our group has successfully used CRISPR to generate multiple *Culex quinquefasciatus* mutants, establishing a platform of validated reagents for future work[31]. While the delivery of transgenes to the *Culex* germline has been achieved via *Hermes* transposable elements[32,33], CRISPR-based transgene delivery has not yet been accomplished within the species.

CRISPR-based gene drives offer tremendous potential for engineering wild populations due to their ability to self-propagate, and bias their inheritance toward super-Mendelian rates (>50%)[34]. These engineered elements consist of Cas9- and guide-RNA (gRNA)-expressing genes, which are integrated at the site that is targeted by the gRNA. When the two components are expressed in the germline, the wild-type allele is cut and converted to a copy of the gene drive by means of the endogenous, homology-directed repair (HDR) pathway[34]. This process increases the frequency at which the gene drive element is passed on to the offspring, allowing it to spread into a target vector population and achieve disease relief by delivering engineered genes[20,22,24,25].

In *Culex*, validated promoters to drive both Cas9 and gRNA expression in vivo are needed, but so far only one research group has analyzed the activity of *Culex quinquefasciatus* U6 promoters in cell culture[35]. Therefore, a complete set of Cas9 and gRNA promoters to drive expression in vivo is required before gene drive development in *Culex* species can be achieved. Another technical barrier for gene drive development lies in the initial HDR-based site-specific delivery of the gene drive transgene into the mosquito genome. HDR-based transgenesis in mosquitoes has been shown to be an inefficient process, requiring the injection of large amounts of eggs and labor-intensive screening to obtain a few positive transformants[20,36]. Given that these techniques have not yet been developed for *Culex*, HDR-based

transgenesis remains a major hurdle to achieve gene drives in these mosquitoes.

Here we develop a set of promoters for the expression of Cas9 and gRNA in *Culex quinquefasciatus* and validated them in vitro and in vivo. We then optimize our constructs with gRNA scaffold variants, employed these constructs to evaluate HDR-based transgenesis in *Culex quinquefasciatus*, and delivered a ~9 kb Cas9-expressing transgene to the *cardinal* locus. Next, we validate the ability of this line to drive expression of Cas9 in the germline, a capability that will be essential for future gene drive development. We further show that the gRNA improvements observed in *Culex quinquefasciatus* translate to the fruit fly, *Drosophila melanogaster*, increasing HDR-based transgenesis in both species and supporting their potential application in other insects with limited CRISPR protocols. Lastly, we show how these gRNA scaffold alterations can be applied to gene drives and boost their efficiency in fruit flies, demonstrating new options for gene drive tuning in other species. Overall, the findings and resources reported here pave the way for developing gene drive strategies and other CRISPR-based technologies for *Culex* mosquitoes population control.

## Results

**Generation and validation of transgenes for the expression of CRISPR components**. To generate plasmid reagents for the efficient expression of Cas9 and gRNA, we identified regulatory regions of *Culex quinquefasciatus gene* orthologs of ones that have been previously used for efficient Cas9/gRNA expression in other species[20,36–38]. For Cas9, we selected the ubiquitously expressed genes *Actin5C* (CPIJ009808) and *Rpl40* (CPIJ002413), as well as the two germline-specific genes *vasa* (CPIJ009286) and *nanos* (CPIJ011551). For gRNA expression, we chose to use regulatory sequences from the small nuclear RNA U6 genes, which are *Pol-III*-transcribed and have been successfully used to drive gRNA expression in species ranging from the fruit fly[38] to humans[39]. We identified seven U6 genes from the published *Culex quinquefasciatus* reference genome: *Cq*-U6:1 (CPIJ039653), *Cq*-U6:2 (CPIJ039728), *Cq*-U6:3 (CPIJ039543), *Cq*-U6:4 (CPIJ039801), *Cq*-U6:5 (CPIJ040819), *Cq*-U6:6 (CPIJ039596), and *Cq*-U6:7 (CPIJ040693)[40]. Out of all the planned constructs, we managed to clone and obtain all U6s promoters except *Cq*-U6:3 and *Cq*-U6:5, and for U6:2 we obtained two versions, *Cq*-U6:2 and *Cq*-U6:2b, from two lines sourced from Alabama and California, respectively (Supplementary Fig. 1). With these generated constructs we then proceeded to validate the functionality of the carried promoters.

**Validation of CRISPR reagents in *Culex quinquefasciatus* ovarian cell line**. The optimal expression of CRISPR-Cas9 system components is key to achieving high editing efficiencies in gene drives and other genome engineering applications. To evaluate our newly built CRISPR reagents, we tested their activity in a controlled in vitro cell culture system: the *Culex quinquefasciatus* Hsu cell line, which is derived from adult ovarian tissue[41]. We transfected Hsu cells with a GFP reporter and different combinations of Cas9- and sgRNA-expressing constructs to evaluate their editing efficiencies in targeting the *kynurenine hydroxylase* (*kh*) (CPIJ017147) locus with the validated *kh3*-gRNA[31].

The workflow for screening and analyzing the editing efficiency of these constructs is indicated in Fig. 1a. Using this workflow, we evaluated the activity of six different *Culex quinquefasciatus* U6 promoters driving *kh3*-gRNA expression (*Cq*-U6:1, *Cq*-U6:2, *Cq*-U6:2b, *Cq*-U6:4, *Cq*-U6:6, *Cq*-U6:7) (Supplementary Fig. 2).

A Cas9-expressing plasmid (*Cq*-*Actin5C* > Cas9) was transfected with each gRNA-expressing plasmid at a fixed molar ratio, and a range of editing efficiencies was observed for each U6

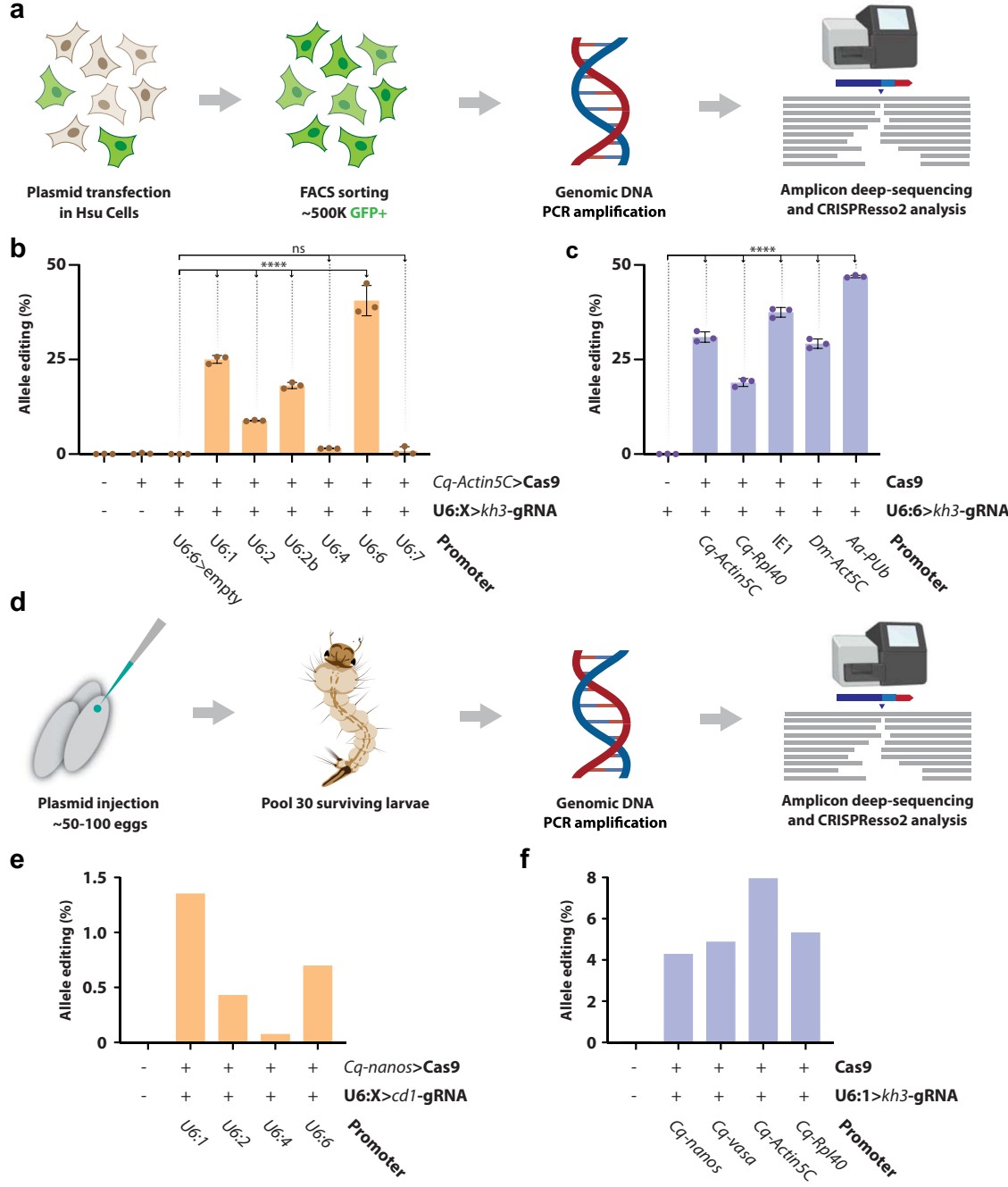

**Fig. 1 Evaluation of gRNA and Cas9 constructs in *Culex quinquefasciatus* cells and developing embryos. a** Schematic of the in vitro workflow for the evaluation of CRISPR-reagent editing efficiency in Hsu cell line: a plasmid mix containing a GFP reporter cassette was co-transfected at day 0; cells were expanded and cultured for 12 days after transfection; GFP-positive cells were sorted via fluorescence-activated cell sorting (FACS) and used to prepare genomic DNA; gRNA target specific region was amplified and deep sequenced; CRISPR editing efficiency was inferred using CRISPResso2 (Supplementary Fig. 2). **b** Histogram representing editing efficiency (%) of *kh3* target locus in Hsu cells when co-transfected with a mixture of *Cq-Actin5C* > Cas9 and various *Culex quinquefasciatus* native U6 promoters expressing *kh3*-gRNA. Statistical comparisons against the empty vector control were generated using a one-way ANOVA followed by Tukey's multiple comparison test (*p* values: U6-1 < 0.0001; U6-2 < 0.0001; U6-2b < 0.0001; U6-4 = 0.9995; U6-6 < 0.0001; U6-7 > 0.9999). **** = $P_{tukey}$ < 0.0001, ns = $P_{tukey}$ > 0.05. **c** Co-transfection of the *Cq-U6:6* > *kh3*-gRNA paired to different Cas9-expressing plasmids. The same statistical analysis as above was performed against the control. All pairwise comparisons to the control are significant (*p* values: *Cq-Actin5C* < 0.0001; *Cq-Rpl40* < 0.0001; *IE1* < 0.0001; *Dm-Act5C* < 0.0001; *Aa-pUb* < 0.0001). Histogram bars represent the mean; error bars and dots represent SD and distribution of three biological replicates. *Cq C. quinquefasciatus, Dm D. melanogaster, Aa Ae. Aegypti*. **d** Schematic of the protocol used to evaluate editing activity of the generated constructs in developing embryos. Each plasmid mixture was injected in freshly laid eggs; hatched larvae were collected at ~48 h; genomic DNA was prepared from larvae pools for PCR; a deep sequencing analysis was performed on the targeted region. **e** Graph displaying the percentage of edited alleles observed by co-injecting the *Cq-nanos* > Cas9 plasmid with different U6 promoters driving the *cd1*-gRNA. **f** Percentage of edited alleles observed when co-injecting the *Cq-U6:1* > *kh3*-gRNA construct with different Cas9 plasmids. See "Methods" and Supplementary information for definitions of allele editing (%) and specifics on plasmid constructs/transfection mixes. **a**, **d** Partially created with the help of BioRender.com.

promoter (Fig. 1b). The *Cq*-U6:2b performed significantly better than the *Cq*-U6:2 (mean = 18.2% and 8.8%, respectively), suggesting the presence of regulatory elements included in the *Cq*-U6:2b construct that are not present in the shorter *Cq*-U6:2 construct. The *Cq*-U6:4 and *Cq*-U6:7 promoters showed the lowest editing activity, which was not significant compared to the controls (mean = 1.55% and 0.7%, respectively). In contrast, *Cq*-U6:1 and *Cq*-U6:6 showed the highest efficiencies, editing 25% and 40.6% of total alleles, respectively, suggesting that these promoters may be the best choice for this system. Interestingly, for the *Cq*-U6:1 promoter we observed a twofold increase in editing efficiency when extending the culturing time after transfection, confirming that a longer exposure to the CRISPR reagents leads to increased genome editing in cells (Supplementary Fig. 3a).

Next, to compare the editing efficiencies of various Cas9 expression constructs, we co-transfected a constant amount of a gRNA-expressing plasmid (*Cq*-U6:6 > *kh3*-gRNA) with one of five copy number-balanced Cas9 constructs under different *Pol-II* ubiquitous promoters. We tested two *Culex quinquefasciatus* native promoters (*Cq-Actin5C*, *Cq-Rpl40*) and three heterologous promoters, derived from the baculovirus immediate early 1 promoter (*IE1*), the *D. melanogaster Actin5C* (*Dm-Actin5C*), and the *Aedes aegypti* poly-ubiquitin (*Aa-PUb*) genes. All Cas9-expressing constructs displayed robust and sustained editing activity significantly higher than the control (Fig. 1c). Among the *Culex quinquefasciatus* native promoters, the editing efficiency for *Cq-Actin5C* > Cas9 and *Cq-Rpl40* > Cas9 was 30.9% and 18.9% in average, respectively. Heterologous expression of Cas9 under *Dm-Actin5C* performed similarly to the native *Cq-Actin5C* promoter (mean = 29.2%). The *IE1* and *Aa-PUb* promoters were the most active, inducing higher editing rates at the *kh* locus (mean = 37.5% and 46.9%, respectively). Overall, the high editing efficiencies observed look promising for the future use of CRISPR-Cas9 in Hsu cells during functional genomic studies and expand the CRISPR tool-set for this mosquito species.

**In vivo validation of CRISPR reagents in *Culex quinque-fasciatus*.** After validating the *Culex quinquefasciatus* CRISPR reagents in a cell culture system, we next evaluated the activity of these constructs in vivo. First, we built *Cq*-U6:1, *Cq*-U6:2, *Cq*-U6:4, and *Cq*-U6:6 gRNA constructs that targeted the *cardinal* locus (CPIJ005949) (U6:X > *cd1*-gRNA) using a validated gRNA[31]. We chose the *cardinal* gene for this analysis for three reasons: (1) we have previously validated editing at this locus using *cd1*-gRNA and built a *cardinal*− homozygous line[31], which we will use later in this work; (2) *cardinal*− homozygous mutants display a visible phenotype, a lighter than wild-type, red eye, which darkens over time and potentially leads to an almost wild-type fitness of an eventual homozygous transgenic line; and (3) we could directly use the reagents validated in embryos for the next step of this project, aiming to obtain HDR-based transgenesis. To proceed with this analysis we co-injected each U6:X > *cd1*-gRNA construct into *Culex quinquefasciatus* embryos along with a germline-specific promoter of Cas9 (*Cq-nanos* > Cas9). We next collected the newly hatched larvae from different batches of injection, extracted DNA to amplify regions around the Cas9 targeted site, and then analyzed these products via deep sequencing (Fig. 1d).

We observed varying degrees of in vivo activity for each U6, as reflected by editing rates observed at the *cd1* site (Fig. 1e). Among each U6, *Cq*-U6:4 showed the lowest editing activities in vivo (0.07%) (Fig. 1e), which was consistent with our in vitro results. In contrast, the remaining U6s exhibited moderate cutting efficiencies (0.43–1.35%) (Fig. 1e). This might be caused by

either limited Cas9 expression levels driven by the germline-specific *nanos* promoter or by low gRNA amounts due to time-limited expression in the developing embryo, as we perform the DNA extraction only 48 h after injection. As the *Cq*-U6:1 showed the highest in vivo editing (Fig. 1e), we chose it as a candidate U6 promoter to drive gRNA expression for all subsequent constructs.

After validating U6 promoter candidates in vivo, we also explored the in vivo activity of multiple *Culex quinquefasciatus* promoters for Cas9. Using the workflow shown in Fig. 1d, we co-injected a *Cq*-U6:1 > *kh3*-gRNA plasmid, targeting the *kh* locus, along with one of four Cas9 plasmids (*Cq-Actin5C* > Cas9, *Cq-Rpl40* > Cas9, *Cq-nanos* > Cas9, and *Cq-vasa* > Cas9). Each native *Culex quinquefasciatus* promoter displayed functional editing activity with varying efficiency (Fig. 1f). The ubiquitous *Cq-Actin5C* and *Cq-Rpl40* promoters showed cutting efficiencies of 7.95% and 5.32%, respectively (Fig. 1f), while the germline-specific *Cq-nanos* and *Cq-vasa* promoters returned slightly lower editing rates of 4.29% and 4.88% (Fig. 1f). These numbers confirm that all *Culex quinquefasciatus* promoters tested are capable of driving Cas9 expression in vivo. Overall, we successfully validated our CRISPR reagents in vivo and confirmed the prior activity observed within Hsu cells. This suggests that these promoters may serve as valuable tools for future CRISPR approaches in *Culex*.

**Generation of a Cas9-expressing line by site-directed trans-genesis in *Culex quinquefasciatus*.** To evaluate HDR-mediated transgenesis in *Culex quinquefasciatus*, we employed our CRISPR reagents to target the insertion of the *vasa*-Cas9 transgene into the *cardinal* locus using the *cd1*-gRNA from our previous work[31]. To further increase our odds of recovering transformants, we built a single plasmid that functions both as an HDR template as well as the source of Cas9 and gRNA elements. This plasmid comprises three main components: (1) the transgene to be inserted (including the *vasa*-Cas9 cassette and an Opie2>DsRed marker) flanked by; (2) two ~1.5 kb homology arms (HAs) matching the genomic sequences of the *cardinal* locus abutting the *cd1*-gRNA target site; and (3) a gRNA cassette outside the HAs to produce the gRNA necessary for the targeted insertion (Fig. 2a).

We also explored the use of gRNA scaffold variants that have been previously shown to increase knockout efficiency in human cells[42], and built two additional constructs carrying these gRNA scaffold alterations: (1) an extended loop with an additional five base pairs (bp) that better resemble the system's native state, hereon termed "Loop"; and (2) the same "Loop" alteration supplemented by the $T_4 > C$ mutation, a single nucleotide change in a stretch of four Ts at the beginning of the gRNA scaffold, which otherwise can be interpreted as a stop signal by the RNA-Polymerase III, hereon termed "Loop + Mutation" (Fig. 2b, FASTA sequences available in the Supplementary information).

To evaluate the efficiency of these constructs and generate a targeted insertion of the Cas9 transgene, we injected each of the plasmids into eggs of a *Culex quinquefasciatus* line derived from our laboratory strain by making the *cardinal* locus isogenic (see methods, Supplementary Fig. 4). The injected G0s were then separated into male and female pools and crossed to our previously described homozygous mutant *cd*−/*cd*− mutant line (Fig. 2c)[31]. The resulting G1s were phenotypically screened: a reddish eye (*cd*−) would indicate a cutting event at the *cardinal* locus, while expression of the DsRed marker would indicate the successful HDR-mediated insertion of the Cas9 transgene (Fig. 2c). While we observed editing in all three conditions, we were able to recover transgenic animals only when using the "Loop" and "Loop + Mutation" constructs, suggesting that these

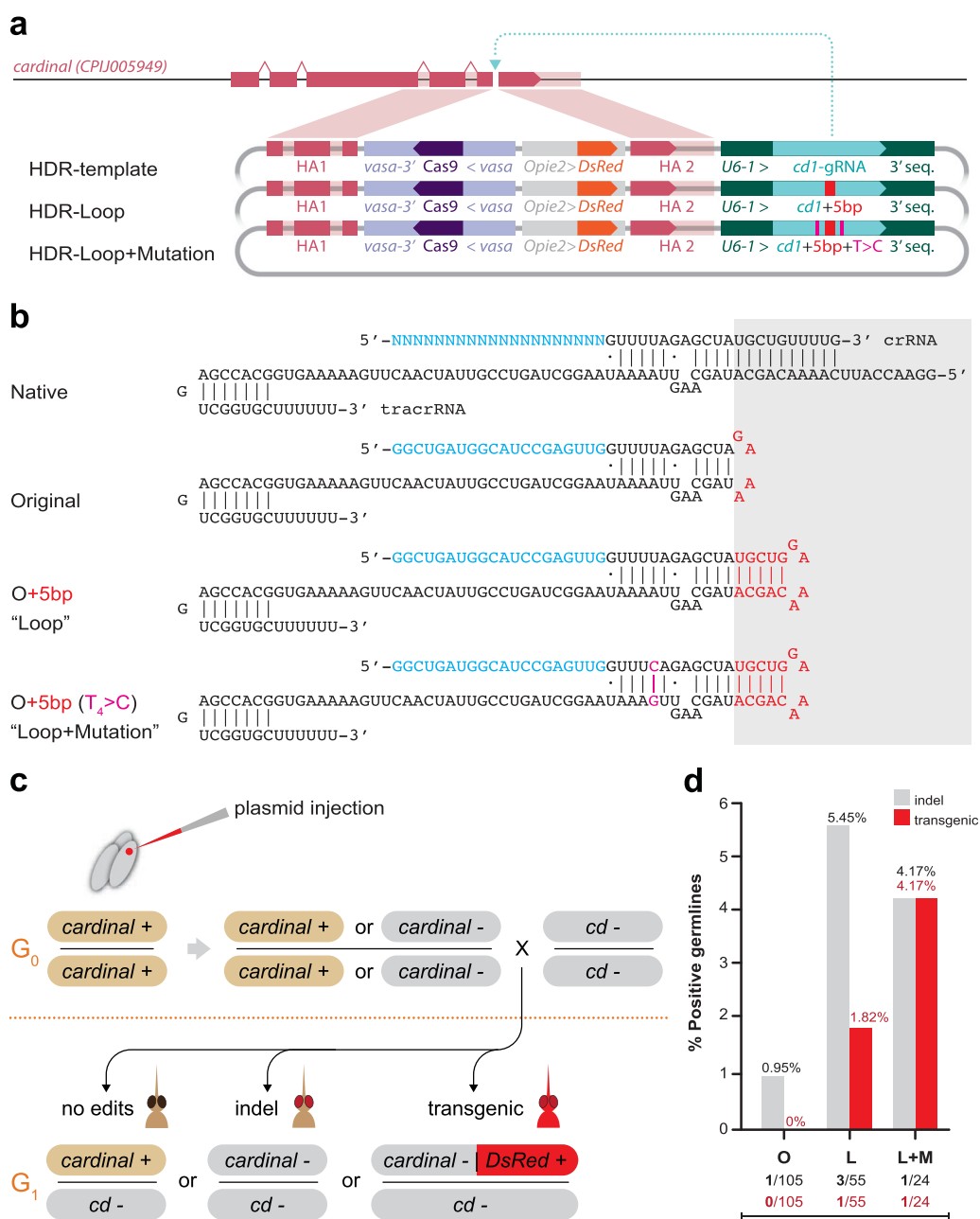

**Fig. 2 Site-directed transgenesis of the _vasa_-Cas9 cassette. a** Three constructs generated for transgenesis in _Culex quinquefasciatus_. Pink shading highlights the location of the homology arms used for site-specific targeting of the _Cq-vasa_ > Cas9 and Opie2 > DsRed transgenes. An additional _Cq-U6:1_ > _cd1_-gRNA transgene is present beyond the right homology arm, preventing its insertion. The representation of the genetic elements is not to scale. **b** Overview of the gRNA scaffold variants used in this study compared to the native fold of the crRNA/tracrRNA pair. The gray shaded area in the figure highlights the synthetic portions of the gRNA variants that were introduced to link the crRNA and the tracrRNA. Red indicated the synthetic loop. Purple indicates the mutation introduced in the gRNA scaffold. **c** Injection and cross scheme used to simultaneously evaluate cutting and transgenesis efficiency of the injected plasmids. **d** Bar graph representing the cutting and transgenesis rates observed in our experimental conditions. The fraction of germlines showing editing (black) or transgenesis (red) over the total germline sampled (_n_) is reported below each condition. O: "Original" gRNA, L: "Loop", L + M: "Loop + Mutation".

gRNA scaffolds could improve transgenesis in _Culex quinque-fasciatus_ (Table 1). Interestingly, the germline cutting efficiency of the template carrying the "Original" gRNA scaffold was lower than the other conditions. The "Original" scaffold showed only 0.95% (1/105 germlines) editing and no transgenesis, compared to 5.45% (3/55) editing and 1.82% (1/55) transgenesis for the "Loop" and 4.17% (1/24) and 4.17% (1/24) for the "Loop + Mutation" construct, respectively (Fig. 2d, Table 1, and Supplementary Data 2). Additionally, when we compare the overall number of cut or transgenic G1 animals obtained using the modified scaffolds to the "Original" scaffold, we observe a significant increase in both editing and transgenesis (_p_ values in Table 1). These combined results support the hypothesis that the two gRNA scaffold variants tested can lead to a substantial increase in cutting activity, in line with previous reports[42]. In turn, a higher cutting rate should promote an increased likelihood

**Table 1 *Culex quinquefasciatus* cutting and transgenesis efficiencies of different HDR templates.**

| HR plasmid | Injected G0 | | | Efficiency (per G0 germline)[a] | | Overall efficiency (out of total G1s)[b] | |
|---|---|---|---|---|---|---|---|
| | Injected eggs | Adult survivors | Survival | Cutting: *cd−/cd−* (%) | Transgenesis DsRed+ (%) | Cutting: *cd−/cd−* (%) | Transgenesis DsRed+ (%) |
| "Original" | 815 | 105 | 12.88% | 1/105 (0.95%) | 0/105 (0%) | 6/5377 (0.11%) | 0/5377 (0%) |
| "Loop" | 1480 | 55 | 3.72% | 3/55 (5.45%) **(p value = 0.055) ns** | 1/55 (1.82%) **(p value = 0.167) ns** | 25/3148 (0.79%) **(p value < 0.0001)\*\*\*\*** | 4/3148 (0.13%) **(p value = 0.014)\*** |
| "Loop + Mutation" | 1151 | 24 | 2.09% | 1/24 (4.17%) **(p value = 0.188) ns** | 1/24 (4.17% **(p value = 0.025)\*** | 22/1126 (1.95%) **(p value <0.0001)\*\*\*\*** | 4/1126 (0.36%) **(p value = 0.0008)\*\*\*** |

A one-tail randomization test for a difference in proportions was performed to determine whether the Loop or the Loop + mutation scaffold variants caused an increase in transgenesis when compared to the "Original" gRNA; the obtained *p* values are reported in the table in bold type for these comparisons for either the "Efficiency per G0 germline" or "Overall efficiency (G1)" along with the significance expressed in using the asterisk convention.
*ns* not significant.
[a]G0 germline cutting and transgenesis efficiencies were calculated as numbers of independent pools that produce either *cd−/cd−* mutant (cutting) or DsRed+ (transgenesis) animals, divided by the total number of crossed G0s. While each pool contains several G0 individuals, our calculations assume only one editing event happened in each positive pool and may underestimate the cutting and transgenesis rates.
[b]The overall cutting and transgenesis efficiencies were calculated as the number of G1 individuals with either *cd−/cd−* (cutting) or DsRed+ (transgenesis) phenotypes divided by the total number of G1s.

of obtaining HDR events in transgenesis efforts. Separately, we tested the same scaffold modifications in our cell-based and embryo systems, however, we did not observe the same trend in editing efficiency (Supplementary Fig. 3).

During our transgenesis efforts, we also evaluated whether the amount of Cas9 provided by the injected plasmid was a limiting factor, and we supplemented the HDR templates with either Cas9 protein, a Cas9 plasmid mixture, or a combination of both. While in most cases we did not observe any cutting events, one injection using a combination of the "Loop" variant with both Cas9 protein and the Cas9 plasmids mixture resulted in cutting within 3/3 pools and transgenesis in 2/3 pools containing a total of 9 G0 mosquitoes (Table S1 and Supplementary Data 2). In a replicate experiment of this condition we were only able to observe cutting, suggesting that either stochasticity was involved in the discrepancy between replicates, or other factors beyond the altered variables, such as needle quality, injection mix deposits, or developmental timing of the eggs used for injection. While these observations are inconclusive, the additional supplement of Cas9 sources has the potential to further increase transgenesis efficiency in this mosquito and warrants further exploration.

**Validation of the *Culex quinquefasciatus vasa*-Cas9 transgenic line**. To establish a homozygous *vasa*-Cas9 transgenic line for future studies, we mated four transgenic males recovered from the previous injections to wild-type females. Only two such crosses were successful, and from each cross' offspring, mosquitoes with a genotype of $cd^{[vasa\text{-}Cas9,\ DsRed+]}/cd+$ were then intercrossed to obtain homozygous animals in the following generation, and establish transgenic lines. As expected, animals homozygous for the transgene displayed the *cardinal−* phenotype (Fig. 3a, b) as well as the ubiquitous DsRed fluorescence in both larvae and adults (Fig. 3a', b'). Interestingly, the strong expression of the Opie2-DsRed transgene is visible to the naked eye as an orange–pink coloration of the larvae (Fig. 3a), which could allow the Cas9 transgene to be tracked with a regular dissecting microscope in the absence of a fluorescence setup.

To confirm the correct insertion of the transgene, we designed oligonucleotide primers to amplify four overlapping fragments spanning the entire transgene (Fig. 3c). Indeed, we were able to amplify segments covering the entire region and show that the amplicons corresponded to the expected size (Fig. 3d). Accurate and mutation-free transgene insertion was further confirmed via sanger sequencing.

After validating transgene insertion within this line, we next tested whether our *vasa*-Cas9 transgene is able to drive expression of Cas9 protein in the germline. We performed a functional evaluation of transgene activity by injecting transgenic eggs with a gRNA source that targets the *kh* gene, causing a white-eye phenotype when disrupted[28,31]. We injected eggs from a heterozygous Cas9 line with an in vitro transcribed (IVT) or a plasmid-expressed *kh3*-gRNA previously validated within our lab[31]. Since the eggs were obtained from a heterozygous line, we screened the surviving adults for the presence of the DsRed marker and discarded any animals without fluorescence prior to crosses with a homozygous *kh−/kh−* mutant line (Fig. 3e)[31]. This cross was performed to evaluate the occurrence of genome editing events happening in the germline of G0 injected animals. Successful edits in the G0 germline at the *kh* locus (*kh\**, Fig. 3e), would combine with *kh−* alleles provided by the homozygous *kh−/kh−* line, and lead to white-eye phenotype in the G1 (*kh\*/kh−*, Fig. 3e). We then screened the resulting G1 offspring for presence of the *kh−* white-eye phenotype, which would indicate a successful Cas9/*kh3*-gRNA-driven *kh+* allele disruption in the G0 germline. We recovered G1 *white*-eye animals in both conditions tested, and molecularly validated the editing in these animals by sequencing PCR amplicons (Supplementary Fig. 5). While the IVT-gRNA yielded ~50% editing efficiency, the injection of the plasmid gRNA resulted in the recovery of ~1% of mutant G1s (Fig. 3e and Supplementary Data 3). It is possible that Cas9 protein preloaded in the egg, can readily bind the injected IVT-gRNA leading to early, and efficient editing, while the plasmid could result in a more gradual production of the gRNA leading to lower editing in the germline. While these two strategies are not directly comparable due to the differences in gRNA delivery, both results confirm that the *vasa* promoter in our transgene is capable of driving efficient expression of Cas9 in the germline and producing genome edits at the *kh* locus.

**gRNA scaffold variants improve transgenesis in *Drosophila melanogaster*.** Since our *Culex quinquefasciatus* transgenesis data showed that the tested gRNA scaffold variants could increase both cutting and transgenesis rates, we wondered whether this observation would apply to other laboratory models. To test this hypothesis, we evaluated the ability of the "Loop" scaffold variant to improve transgenesis efficiency in *Drosophila melanogaster*. Unlike our mosquito transgenesis experiment, which required pool-mating for efficient recovery of offspring, the fruit fly would instead allow us to perform single-pair crosses and generate a more robust evaluation of transgenesis efficiencies.

To test this hypothesis, we generated two different HDR templates to test targeted transgene integration in the fruit fly, which were designed to target the insertion of a gRNA cassette and a GFP marker at the *white* (*w*) locus. These templates consisted of three elements: (1) a *w5*-gRNA expressed under the U6:3 *Drosophila melanogaster* promoter with either the "Original" or "Loop" scaffold; (2) a GFP fluorescent marker under the control of the 3xP3 promoter within the eye; and (3) two HAs flanking the *w5*-target cut site to drive transgene integration (Fig. 4a). To test the transgenesis efficiency of these constructs, we

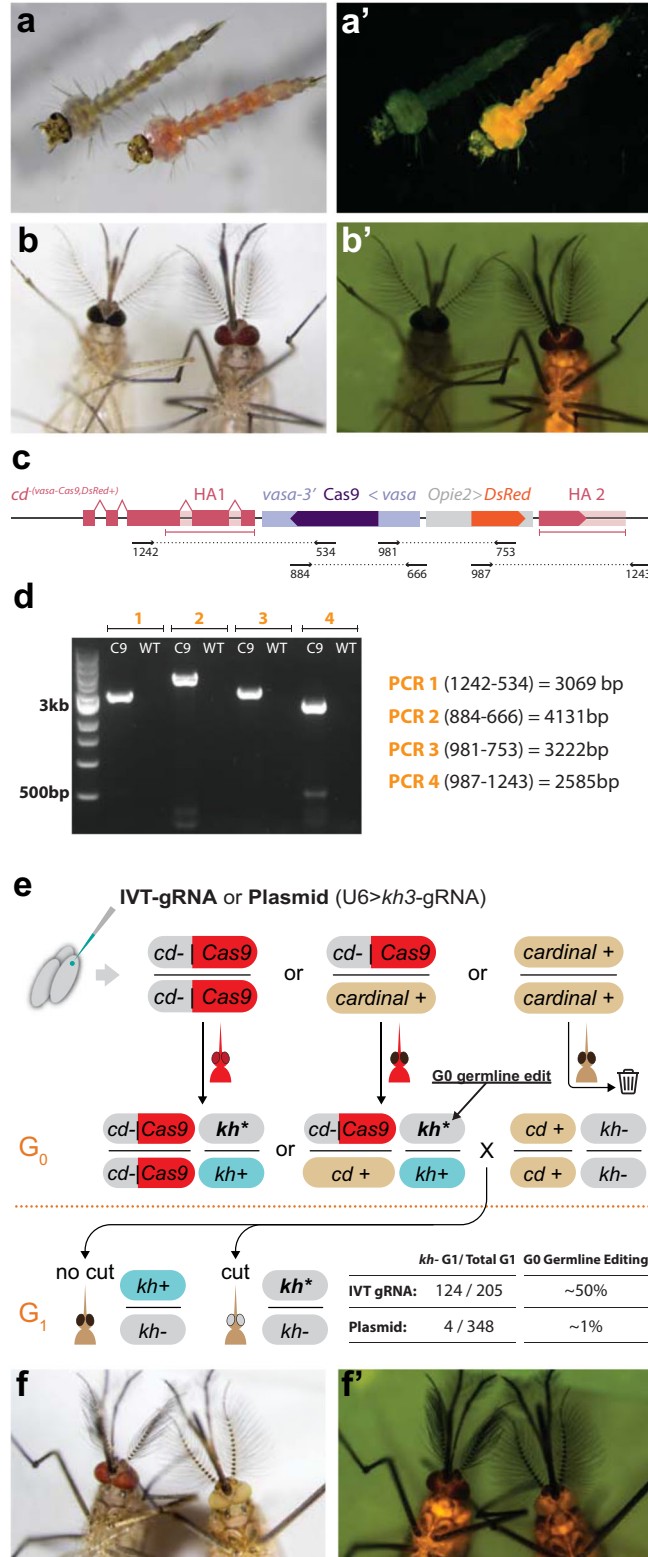

**Fig. 3 Phenotypic, molecular, and functional validation of the *vasa*-Cas9 transgene. a, b'** Phenotypes of homozygous *vasa*-Cas9 (right) compared to wild-type (left). Individuals display the *cardinal*− eye phenotype and the DsRed fluorescence marking the transgene in both (**a, a'**) larvae and (**b, b'**) adults. **a, b** Brightfield image. Note how the expression of the transgene is visible as pink pigmentation, observable to the naked eye. **a', b'** DsRed fluorescence filter. **c** Representation of the transgene inserted within the *cardinal* open reading frame and the primers used to generate the four diagnostic amplicons. **d** Gel electrophoresis of the diagnostic amplicons. C9 Cas9, WT wild-type. **e** Experimental outline of the functional validation of the Cas9 transgenic by injection of eggs with either in vitro transcribed (IVT) *kh3*-gRNA or a plasmid expressing *kh3*-gRNA. Cas9-positive animals were crossed to a *kh*− line to evaluate gRNA activity in the G1. The editing rate observed for either IVT-gRNA or plasmid is reported. *kh*+ wild-type *kh* allele, *kh*− mutant *kh* allele coming from the homozygous *kh*−/*kh*− line, *kh** mutant *kh* allele generated in the G0 germline by Cas9-directed cleavage. A control male from the injected Cas9 line (left) compared to a DsRed+, *kh*− animal (right), in (**f**) brightfield and (**f'**) red fluorescence channel.

11/42 (26%) of transformants with the original gRNA plasmid (one-tail randomization test for a difference in proportions, *p* value = 0.039, Fig. 4c and Supplementary Data 4). Additionally, when evaluating the overall recovery of G1 transgenic animals, we observe a significant increase in the fraction of GFP + G1 flies recovered when using a construct carrying the "Loop" variant (279/2379) in comparison to the "Original" gRNA (185/1982) (one-tail randomization test for a difference in proportions, *p* value = 0.0053, Table 2). These combined results show that the "Loop" gRNA scaffold variant can improve transgenesis in a second organism, suggesting that this simple gRNA modification could benefit transgenesis efforts in other species without established protocols.

**gRNA scaffold variants improve gene drive efficiency in *Drosophila melanogaster*.** Since we observed improved activity when using scaffold variants in both *Culex quinquefasciatus* and *Drosophila melanogaster*, we wondered whether these variants could be effectively used in a gene drive strategy, and whether they would improve efficiency in these genetic systems. To test these hypotheses, we employed a split-gene drive strategy (aka Copy-Cat) in *Drosophila*[19,43,45,46]. Briefly, a CopyCat element consists of a transgene containing a gRNA element (and a GFP marker) flanked by two HAs, which match the targeted locus sequence on each side of a cut site generated by the gRNA within the transgene (Fig. 5a). Under normal conditions, these constructs behave as regular transgenes, and are inherited in a Mendelian fashion. When flies carrying these transgenes are combined with ones carrying a Cas9 source, the Cas9 protein and the gRNA within the CopyCat cleave the wild-type allele, which is then repaired by HDR, promoting gene drive of the CopyCat element (Fig. 5a). We built four CopyCat constructs with a different gRNA scaffold: (1) the control "Original" gRNA, (2) "Loop"; (3) "Mutation"; and (4) "Loop + Mutation", and we obtained transgenic animals carrying their insertion in the *white* locus (Supplementary Fig. 6). Note that the first two constructs are the same that we employed above, in our evaluation of transgenesis efficiency in the fruit fly; all these constructs were inserted using an integration strategy analogous to what is depicted in Fig. 4a.

To test biased gene drive inheritance using these four lines, we crossed males carrying a DsRed-marked *Dm-vasa* > Cas9 transgene inserted at the *yellow* locus[44] to females containing the *w5*-CopyCat at the *white* locus (Fig. 5b). We then selected F1 virgin females carrying both the Cas9 and the *w5*-CopyCat, and crossed them to *w*−/*w*− mutant males in single-pair crosses. To evaluate

injected each plasmid into a *Drosophila vasa*-Cas9 line, expressing Cas9 in the germline (Fig. 4b)[43,44]. In each case, the injected G0 adults were single-pair crossed to *w*−/*w*− mutant flies, and the presence of the GFP marker in the resulting G1 offspring indicated successful transgene integration. The "Loop" gRNA construct showed a significantly higher germline transformation rate than the original gRNA plasmid, with 21/48 (44%) of individual G0 crosses producing G1 transformants compared to

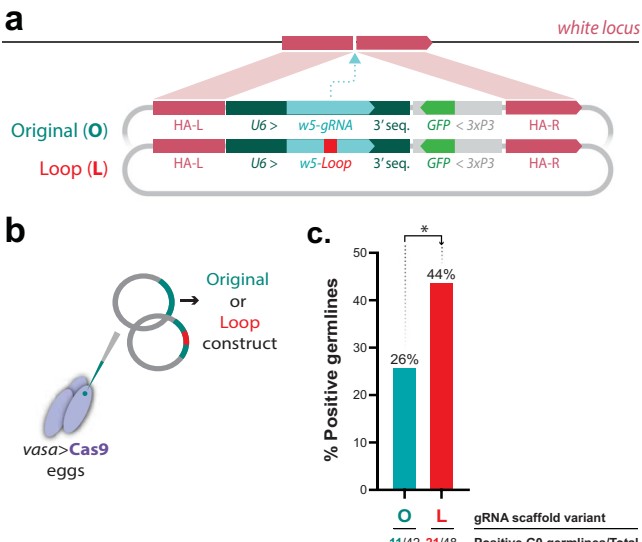

**Fig. 4 Evaluation of transgenesis efficiency in *Drosophila melanogaster* using gRNA scaffold variant. a** Constructs designed for *Drosophila* transgenesis experiments. Both constructs contain the *w*5-gRNA driven by the *Drosophila* U6:3 promoter and are marked with GFP fluorescence to identify positive transformants. Two homology arms flank the gRNA element to facilitate HDR-mediated transgene integration into the *Drosophila* genome. The two constructs differ only in their gRNA scaffold for the presence of either the "Loop" modification or the control, "Original" construct. **b** Both constructs were injected separately into a *Dm-vasa* > Cas9 line to ensure Cas9 expression in the germline. **c** Germline transformation rates were calculated by dividing the number of G0 independent single-pair crosses giving transformants (GFP+) by the total number of G0 crosses performed. A one-tail randomization test for a difference in proportions was performed to determine whether the "Loop" scaffold variant caused an increase in transgenesis when compared to the "Original" gRNA (*p* value = 0.039).

the F1 germline transmission of the CopyCat transgene, we scored eye and fluorescence phenotypes in the resulting F2 progeny (Fig. 5b). Since the Cas9 and the *w*5-CopyCat transgenes are inserted at the *yellow* and *white* loci, respectively (which are closely linked on the X chromosome, at ~1 centiMorgan), this arrangement allowed us to use the Cas9-DsRed transgene as a proxy for the receiver chromosome. This enabled us to disentangle the F2 outcomes using their phenotype and separately track the receiver (Cas9-DsRed) and donor (GFP-only) chromosomes (Fig. 5c). By tracking the DsRed and using a specific genetic cross, we could distinguish events that lead to no cutting, indel generation, and conversion events (Fig. 5c), allowing us to separately evaluate the cutting efficiency (cut chromosome/total receivers) and HDR rate (converted to CopyCat/cut chromosomes) in addition to the overall inheritance of the GFP transgene (Fig. 5d).

Similar to previous studies, we observed super-Mendelian inheritance of the CopyCat constructs inserted in *white*, and our control construct carrying the "Original" gRNA scaffold displayed 63% inheritance of the transgene (Fig. 5d and Supplementary Data 5). The scaffold variants increased inheritance of the transgene up to 82%, 69%, and 71% inheritance for the "Loop", "Mutation", and "Loop + Mutation", respectively. This inheritance data confirm the potential use of these variants to boost gene drive efficiency (Fig. 5d and Supplementary Data 5).

To further evaluate the performance of the different gene drive elements, we quantified the cutting rates for each condition.

Similar to the inheritance data, we observed significantly increased cutting rates for the optimized gRNA variants (77% for "Loop", 53% for "Mutation", and 54% for "Loop + Mutation") compared with the control ("Original" gRNA, 33%) (Fig. 5d, statistical analysis, along with raw data included in Supplementary Data 5). Lastly, we also evaluated the percentage of cut alleles that were successfully converted to a gene drive (HDR conversion efficiency). Despite having a lower overall cutting efficiency, the control displayed a conversion efficiency of 74%, comparable to that observed for the "Mutation" (69%) and "Loop + Mutation" (72%) variants. Remarkably, the "Loop" gRNA variant was able to significantly boost the fraction of cut alleles that were efficiently converted through HDR to 82% (Fig. 5d and Supplementary Data 5). Altogether, these results indicate that the "Loop" modification confers a significantly higher cutting and conversion efficiency compared to the "Original" gRNA and the other conditions tested.

## Discussion

Here we describe a plasmid toolkit for the expression of Cas9 and gRNAs in the mosquito *Culex quinquefasciatus*. This toolkit was functionally validated in cells and in vivo and was optimized via modifications of the gRNA scaffold to successfully generate the first transgenic Cas9 line in *Culex quinquefasciatus*. We further show that these optimized gRNAs can improve both transgenesis and gene drive efficiency in *Drosophila melanogaster*, supporting their potential use for genome editing and gene drive applications in a wide range of insect pests and disease vectors.

Our Cas9 and gRNA expression toolkit displayed varying degrees of activity, making them a potentially valuable resource for applications needing different levels of genome editing. The activity of the *Culex quinquefasciatus* U6 promoters were somewhat consistent with previous work analyzing the binding activity of dead-Cas9/gRNA complexes in cell lines, including Hsu cells[35]. Furthermore, the fact that these constructs showed activity in Hsu cells is promising, as these cells might retain germinal-like features and possibly reproduce gene editing outcomes happening in germinal cells during standard transgenesis or deployment of gene drives.

With the aid of optimized reagents carrying modifications to the gRNA scaffold, we were able to achieve site-directed HDR-mediated transgenesis of a ~9 kb cassette containing a germline-expressed Cas9, and this newly generated *Culex quinquefasciatus* Cas9 line was confirmed to express in the germline. This Cas9 line should be a valuable tool for the researchers investigating this disease vector, and should boost the development of CRISPR-based genetic control strategies in this mosquito, such as gene drives[20,24] or gSIT[21,26]. As for gene drive, CRISPR-based genetic SIT also relies on the combination of a Cas9 and gRNAs expressing transgenes to generate male-sterile-only offspring. The validated reagents that we describe are the first stepping stones toward the development of these technologies in *Culex quinquefasciatus*.

The gRNA scaffold variants used in our experiments did not seem to affect cleaving activity in cells (or embryos), which may be explained by insufficient plasmid expression during the short-window experimental timeline in embryos. In striking contrast, these scaffold variants achieved an increased DNA cleavage in our transgenesis experiments, suggesting the potential use of these variants as options to boost cutting or transgenesis in pioneer insect species and beyond. This notion was supported by our transgenesis experiment in *Drosophila melanogaster*, as our best-performing gRNA scaffold, the "Loop" variant, demonstrated increased recovery in transgenic animals. Previous attempts have been made to evaluate a scaffold variant with a longer loop in

| HDR plasmid | Injected G0 individuals | Efficiency (per G0 germline)[a] | Overall efficiency (out of total G1s)[b] |
|---|---|---|---|
| | G0 single-pair crosses | Vials producing GFP+ (%) | Overall G1 GFP+ recovered (%) |
| "Original" | 41 | 11/41 (26.19%) | 185/1982 (9.5%) |
| "Loop" | 48 | 21/48 (44%) **(p value = 0.039)*** | 279/2379 (13.5%) **(p value = 0.0053)**** |

**Table 2 Drosophila melanogaster transgenesis efficiencies of different HDR templates.**

A one-tail randomization test for a difference in proportions was performed to determine whether the Loop scaffold variant caused an increase in transgenesis when compared to the "Original" gRNA; the obtained p values are reported in the table in bold type for either the "Efficiency per G0 germline" or "Overall efficiency (G1)" along with the significance expressed in using the asterisk convention. ns not significant.
[a]G0 germline transgenesis efficiency was calculated dividing the number of independent G0s that produced G1 GFP+ transgenic animals by the total number of G0 crosses.
[b]The overall transgenesis efficiency was calculated dividing the number of total GFP+ G1 transgenic flies by the total number of G1 scored animals.

Drosophila[47], however, this work did not detect improved mutagenesis rates. It is possible that the loop used in the previous study, which is 5 bp longer than our "Loop" and introduces a second stretch of four thymines, could be read as a stop signal by the RNA-Pol-III (Supplementary Fig. 6b). This would lead to the production of lower gRNA levels, potentially counteracting the effects of the potentially better-performing scaffold in the transgenesis readout.

Lastly, as our research focuses on the future development of gene drives in this vector, we confirmed that the gRNA scaffold modifications can be used as part of a gene drive strategy. We showed that these variants boosted the performance of our gene drive system, and that the "Loop" gRNA variant increased cutting rates and the efficiency at which cleaved alleles are converted by HDR. Interestingly, the combination of the "Loop" and the "Mutation" gRNA modifications did not synergistically boost cutting or conversion efficiencies, as the "Loop + Mutation" inheritance values were comparable to the "Mutation" only. At the same time adding the "Mutation" to the "Loop" variant seems to be detrimental, as the performance of the "Loop + Mutation" gRNA was lower than the "Loop" variant. We suspect that a potential base-pairing between "Loop" and "Mutation" sequences could influence the stability of the gRNA or impact binding of the Cas9 protein, potentially explaining why the "Loop" gRNA performs better in comparison to the "Loop + Mutation" (Supplementary Fig. 6a). Modification of the fourth T to another base within this mutation could possibly enhance the activity of the "Loop + Mutation" variant, and should be investigated in future studies.

This work takes an initial step toward the development of gene drive in Culex quinquefasciatus. Additionally, gRNA scaffold variants improving editing and HDR conversion rates, such as the "Loop" tested here, could be further used to both boost gene drive efficiency and mitigate the generation of resistant alleles in a broad range of vector species[48,49]. As a start, these modifications could be implemented in other mosquito species with more advanced gene drive development, such as Anopheles[20,22,23,25], to deliver even better performance. Lastly, the employment of these gRNA variants could also benefit non-insect model systems for which gene drives are not as efficient, such as the mouse[37], contributing to the development of these technologies in a broader range of organisms.

## Methods

**Cell culture and transfection.** Culex quinquefasciatus Hsu cell line was kindly provided to the Perrimon Lab from Dr. Nelson Lau at Boston University, Boston MA. Hsu cells were maintained at 25 °C in Schneider's medium (Gibco #21720024), 1x MEM NEAA (Gibco #11140050), 1x Penicillin-Streptomycin (Gibco #15140148), and 10% heat inactivated Fetal Bovine Serum (Gibco #16140071). Plasmid DNA for cell transfections was prepared using Qiagen Miniprep Kit and quality was assessed by spectrophotometry and electrophoresis on agarose gel. Plasmid DNA concentration was measured by fluorometric quantification (Qubit, Thermo Fisher) and copy number was normalized between samples in the same experimental condition according to the M.W. of each plasmid and using pUC19 (Thermo Fisher) as DNA "stuffer". The ratio (%) and composition of each plasmid used in each co-transfection mixture was as follows: for U6 comparison experiments, the transfections were performed with a plasmid mixture containing 225 ng (75%) of Cq-Actin5C > Cas9|IE1 > GFP vector as a source of Cas9 and GFP, and 75 ng (25%) of one out of six different copy number balanced gRNA expression plasmids; for Cas9 promoter comparison experiments the mixture contained 60 ng (20%) of Cq-U6:6 > kh3-gRNA vector as a source of gRNA, 225 ng (75%) of one of five different copy number balanced Cas9-expressing vectors and 15 ng (5%) of Ae-PUb > EGFP-NLS as a source of GFP reporter. In all experiments the plasmid mixture included an equal amount of a GFP-expressing plasmid to allow for subsequent fluorescence-activated cell sorting (FACS) of the transfected cells. All experiments conducted in cells were performed using kh3-gRNA-expressing plasmids, except for one of the negative controls that used a Cq-U6:6: "empty" vector. This empty vector construct could also be referenced as expressing a "Non-Targeting" gRNA, as it expresses a gRNA encoding the sequence of the BbsI cloning cassette. Additional information on plasmid vectors and transfection mixtures is specified in Supplementary Fig. 1 and Supplementary Data 1.

To perform transfections, cells were detached from growing flasks using Accumax (Innovative Cell Technologies, Inc) and seeded onto 12-well plates. After 16–24 h (~70% confluency) cells were transfected with 300 ng of plasmid mix using Effectene (Qiagen) and following manufacturer instructions for adherent cell cultures. In total, 24 h after transfection, cells from each well were expanded subsequently to T25 and T75 flasks and cultured for 12 days, or for 18 days after transfection (Supplementary Fig. 2a). At the endpoint, cells were slow-frozen at −80 °C in culturing media supplemented with 10% DMSO v/v, and then preserved until FACS sorting.

**Fluorescence-activated cell sorting (FACS) of transfected cells.** Cell sorting was performed in a Sony MA900 Multi-Application Cell Sorter (Department of Genetics, Harvard Medical School) using a 100-µm sorting chip and average flow rate of 7000 events/s in semi-purity modality (~95 purity). Gates for cell singlets were defined based on forward and back scattering and a subordinate sorting gate was defined based on un-transfected control cells to sort GFP-expressing cells. At the moment of sorting, stored cells were thawed quickly in a 30 °C water bath, washed twice in PBS, and resuspended in fresh culture media. For each sample, 5 × 10^5 GFP-positive cells were sorted directly in a 15 ml falcon tube containing media, and refrigerated at 4 °C for the length of sorting. The sorted cells were pelleted and stored at −80 °C until genomic DNA extraction.

**Preparation of genomic DNA, high-throughput sequencing and analysis.** Genomic DNA was extracted from frozen cell pellets using 1 ml of DNAzol (Thermo Fisher) following the manufacturer's protocol. DNA was ethanol precipitated with addition of GlycoBlue co-precipitant (Thermo Fisher) and resuspended in 30 µl of water. The genomic region around the kh3-gRNA location was amplified from genomic DNA samples and submitted for next generation sequencing (NGS) at the CCIB DNA core (Massachusetts General Hospital, Boston, MA). Primers used for PCR are listed in the Supplementary Methods. The 202 bp PCR product was purified by electrophoresis in agarose gel using a QIAquick Gel Extraction Kit (Qiagen), and eluted in 43 µl of water. Purified DNA amplicon concentration was assessed (Qubit, Thermo Fisher) to ensure sample requirements were met for NGS, and amplicons were sequenced on an Illumina MiSeq instrument using V2 chemistry according to established facility's protocols. Sequencing reads (150 bp paired reads) were demultiplexed, trimmed, and fastq files were used as input for CRISPR editing analysis using CRISPResso2[50]. All samples were analyzed using the batch modality and the same parameters for all experiments performed in cells and embryos (Supplementary Data 1), and allele editing quantification was calculated as: allele editing% [# of reads with modified alleles excluding alleles modified by only substitutions] ÷ [# of total reads aligned to target]. To increase the specificity of our analysis, we did not include mutated alleles modified by only substitutions, as these are mostly derived from recurring SNPs and amplification/sequencing artifacts. Instead, we prioritized the analysis of indels, which are a robust signature of cellular repair mechanisms resulting from Cas9 nuclease activity.

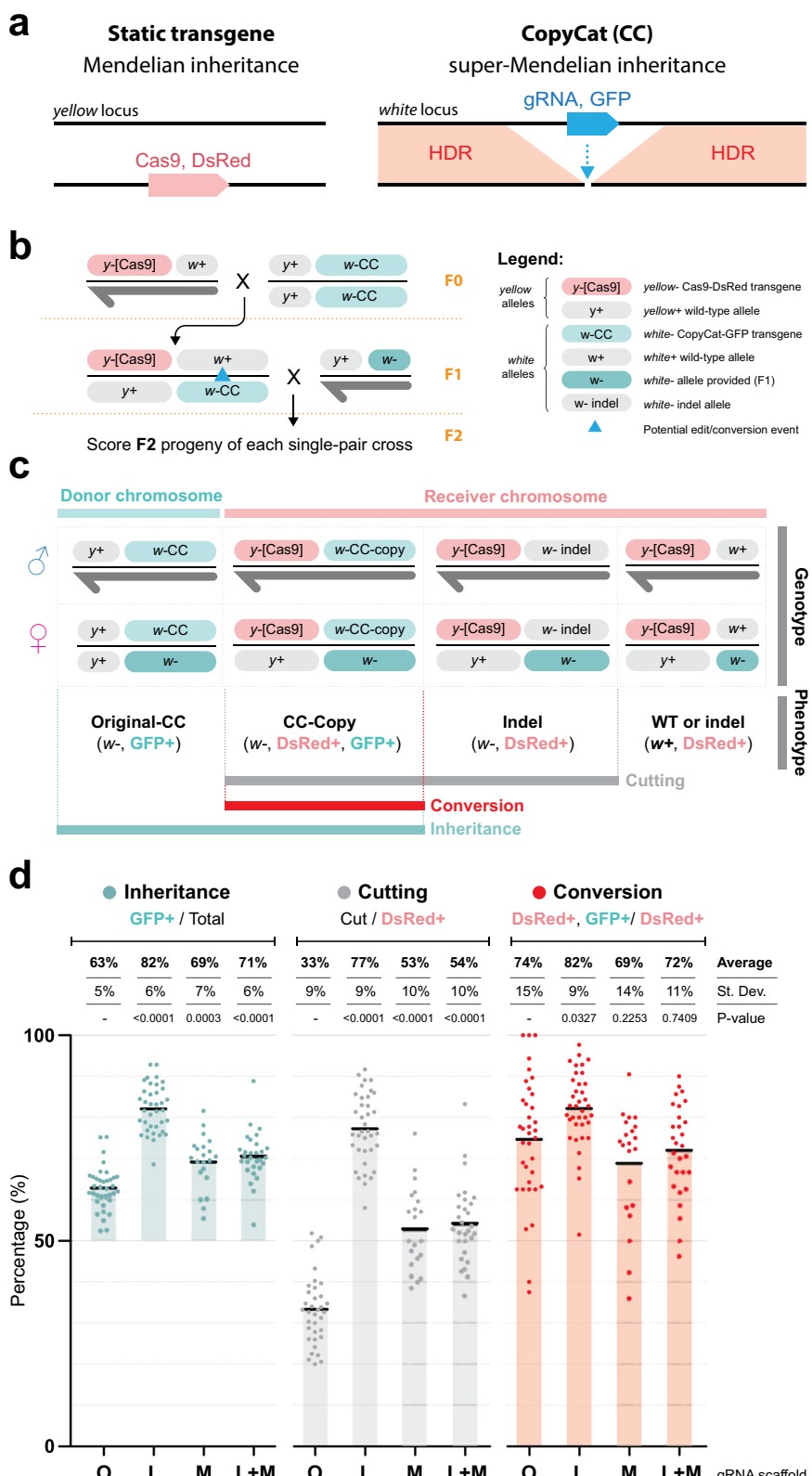

**Mosquito rearing and maintenance for experiments**. The *Culex quinque-fasciatus* (California) strain was kindly provided by Anton Cornell (UC Davis), which was originally collected near the city of Merced, California in the 1950s. The *Culex quinquefasciatus* (Alabama) strain was kindly provided by Nannan Liu (Auburn University), which was collected from Huntsville, Alabama in 2002[16]. The mosquitos were reared at 27 ± 1 °C, 75% humidity, and a 12 h light/dark cycle in the insectary room at the University of California, San Diego. The adults were fed

with 10% sugar water. After mating, females were fed with defibrinated chicken blood (Colorado Serum Company, # 31142) using the Hemotek blood-feeding system. Egg rafts were collected 4 days after blood feeding. Larvae were fed with fish food floating pellets (Blue Ridge Fish Hatchery, USA). Mosquitos were examined and scored with a Leica M165 FC Stereo microscope with fluorescence. All the work presented here followed procedures and protocols approved by the Institutional Biosafety Committee from the University of California, San Diego,

**Fig. 5 Modified gRNA scaffolds improve gene drive in *Drosophila melanogaster*. a** Schematic of the CopyCat (CC) gene drive system. The DsRed-marked Cas9 is a static transgene that is inherited in a Mendelian manner and provides the Cas9 source that allows the GFP-marked CopyCat element to copy itself into the opposing chromosome. **b** Cross schemes of males expressing Cas9 were crossed to virgin females carrying the *w5*-gRNA CopyCat. Collected virgin females (Cas9-DsRed and gRNA-GFP) were crossed to wild-type males to score F2 progeny. The blue triangle in the F1 female represents the germline allelic conversion happening in her germline, which leads to the biased inheritance of the CopyCat in the F2, tracked using the GFP marker. A dark-gray half arrow indicates the male Y chromosome. ♀ = female, ♂ = male. **c** F2 individual outcomes indicating phenotype and genotypes that allow the measurement of inheritance, cutting, and allelic conversion rates. **d** Assessment of inheritance rates (blue dots), cutting rates (gray dots), and allelic conversion efficiency (red dots) for the tested CopyCat constructs. Shaded bars indicate the divergence of the observed average value from the expected one in absence of activity. Black bars indicate the average. Average, standard deviation (St. Dev), and *p* values are reported over conditions compared to the control. *p* values are calculated using a one-way ANOVA test, see the "Methods" and Supplementary Data 5 for more details. ns not significant. O: "Original" gRNA, L: "Loop", M: "Mutation", L + M: "Loop + Mutation".

complying with all relevant ethical regulations for animal testing and research. All maintenance and experiments were performed in a high-security Arthropod Containment Level 2 (ACL2) barrier facility. The wastewater and used containers were disposed of by freezing for 48 h, and subsequently discarded as biohazardous materials.

**Generation of a *Culex quinquefasciatus* line isogenic at the cardinal locus**. We amplified the *cardinal* gene (CPIJ005949) from the California *Culex quinquefasciatus* strain, and two different alleles (named Allele A and Allele B) were identified for this gene consisting of multiple polymorphisms and deletion or insertion around the intended cutting/insertion site. This situation increased the difficulty of obtaining HDR transgenesis. To fix this problem, we collected ten single pupae from the wild-type line (six females and four males) and hatched them individually using the equipment designed by our lab (Supplementary Fig. 4). After hatching, the pupae cases from each single mosquito were collected and their genomic DNA was extracted using a single-fly DNA extraction protocol described by Gloor et al.[51]. PCR amplification of the *cardinal* gene was performed for each individual pupal case with primers listed in the Supplementary Methods, and analyzed by Sanger sequencing. We then combined two females and three males with homozygous alleles (A/A allele) and built a homozygous wild-type line at the *cardinal* locus for later transgenesis experiments.

**Plasmid construction**. Standard molecular biological techniques were applied to generate all constructs analyzed in this work. The genomic DNA of ~30 adult wild-type *Culex. quinquefasciatus* was extracted using DNeasy Blood & Tissue Kit (Qiagen, # 69504). For *Culex* promoters used in this study, their 5′ regulatory and 3′ UTR sequences were amplified with Q5 Hot Start High-Fidelity 2X Master Mix (New England, USA, # M0494S), and cloned on each respective side of either a Cas9 gene or a gRNA preceded by a double-*BbsI* (or in one case *BsmbI*) restriction site linker for later insertion of different gRNAs. Plasmids were built by Gibson Assembly using NEBuilder Hifi DNA Assembly Master Mix (New England Biolabs, # E2621). After assembling, the plasmids were transformed into NEB 5-alpha Electrocompetent Competent *E. coli* (New England Biolabs, # C2989). Correct clones were subsequently confirmed by restriction digestion and Sanger sequencing. All primers used to build plasmids generated in this work are listed in Supplementary Table 2.

**Embryo microinjection**. The injected DNA plasmids were prepared using Pure-Link Expi Endotoxin-Free Maxi Plasmid Purification Kit (Invitrogen, Cat.# A31231), aliquoted based on the concentrations outlined in the manuscripts, and later stored at −80 °C before proceeding to microinjection. All injections were performed on a microinjection station equipped with a FemtoJet 4 microinjector (Eppendorf). The prepared Cas9/sgRNA mixtures were injected into the posterior end of *Culex quinquefasciatus* embryos eggs freshly collected after oviposition (~1 h) to ensure efficient targeting of the germline.

**In vivo activities of different Cas9 promoters, U6 promoters, or gRNA scaffold variants**. To test in vivo activities of different Cas9 promoters, an injection mixture containing 200 ng/µl of *Cq*-U6:1 > *kh3*-gRNA and 200 ng/µl of different Cas9 constructs (*Cq-vasa* > Cas9, *Cq-nanos* > Cas9, *Cq-Actin5C* > Cas9, and *Cq-Rpl40* > Cas9) were prepared for microinjection. For in vivo activities of U6 promoters, a 300 ng/µl solution of *Cq-nanos* > Cas9 was mixed with 300 ng/µl of different *Culex quinquefasciatus* U6s > *cd1*-gRNA plasmids (with *cardinal*−sgRNA-1 placed in different U6 constructs), respectively. All plasmid constructs can be found in Supplementary Fig. 1. The prepared mixture was later injected in *Culex quinquefasciatus* embryos; later 30 freshly hatched larvae were collected for DNA preparation using DNeasy Blood & Tissue Kit (Qiagen, # 69504). For a study of in vivo activity of different gRNA scaffold variants, a mixture containing 150 ng/µl of *Cq-Actin5C* > Cas9 plasmid and a 100 ng/µl of different *Cq*-U6:1 > *cd1*-gRNA scaffold modified variants ("Original", "Loop", and "Loop + Mutation") was co-injected into the embryos of our isogenized (at the *cardinal* locus) *Culex quinquefasciatus* line, and 100 embryos were collected 36-h post injection for DNA

preparation. With prepared DNA samples for each experiment, <500 bp PCRs were performed around *cd*1 or *kh*3 cutting sites with amplicon primers (Supplementary Table 2). The PCR products were later purified with the Monarch Purification Kit (New England Biolabs, #T1030S) prior to deep sequencing analysis. All primers used for PCR amplification, plasmid construction, and deep sequencing are listed in the primer list in Supplementary Table 2. Deep sequencing results (250 bp paired reads, Illumina MiSeq, from Genewiz, inc.) were analyzed by CRISPResso2[50].

**The HDR-mediated transgenesis in *Culex quinquefasciatus***. Different HDR constructs with modified gRNA structures were built (Fig. 2b). Two rounds of injections for each HDR template were performed with a 300 ng/µl solution of HDR plasmid injected in our *Culex quinquefasciatus* line isogenic for the *cardinal* locus. Other conditions were tested by injecting "Loop" modified HDR template supplemented with either 100 ng/µl of recombinant Cas9 protein (DNA Bio Inc., #CP01), 50 ng/µl of Cas9 plasmid mixtures (containing *IE1* > Cas9, *Cq-Rpl40* > Cas9 and *Cq-Actin5C* > Cas9), or both. The injected G0s were divided by sex into two different pools, and were combined with *cardinal* mutant individuals of the opposite sex that were generated in a previous study at a ratio of 3:1–5:1[31]. After mating and blood feeding, egg rafts were collected from each pool. For pools from injected male G0s, all rafts were pooled in the same tray for hatching and counting. While for pools from injected female G0s, each egg raft was separated and hatched individually in trays to record the female G0 germline numbers scored (usually 1 egg raft came from 1 single female). The 3rd instar larvae of G1 were screened and counted for *cd*−/*cd*− mutants and DsRed fluorescent under a Leica M165 FC Stereo microscope with fluorescence. The cutting events were indicated by a *cd*−/*cd*− phenotype and a DsRed marker suggested the successful integration of transgenes.

**Drosophila transgenic line generation and genotyping**. For gene drive experiments (Fig. 5), all constructs were inserted into the same Oregon-R (OrR) strain to maintain a homogeneous genetic background. The Cas9 line was inserted at the *yellow* gene coding sequence, a construct that we validated previously[43]. The CopyCat elements, flanked by specific homology arms (HAs) and marked with GFP, were inserted into the *white* gene coding sequences. For the establishment of all transgenic lines, we received injected G0 flies in the larval stage (~100 larvae) from a commercial *Drosophila* embryo-injection service provider (Rainbow Transgenic Flies, Inc.). Once the larvae hatched as adults, we distributed all G0 individuals into different tubes (4–5 females crossed to 4–5 males). Next, the eyes of the G1 progeny were screened for the presence of the GFP marker, which was indicative of transgene insertion. G1 individuals that were positive for the fluorescent marker were crossed individually to OrR flies (same strain used for injection). To generate homozygous stocks for each transgenic line, G2 flies with the GFP marker were intercrossed, and G3 flies displaying both the fluorescence and the expected white-eye phenotype were pooled. Correct integration of the transgene was confirmed by PCR amplification, as well as through sequencing of the whole construct using primers landing at the genomic region outside of the HAs.

**Drosophila transgenesis experiments**. To compare transgenesis efficiency at the *white* gene in *Drosophila* using the "Loop" modification versus the original construct (Fig. 4), each construct was separately injected into a *vasa*-Cas9 line generated in our laboratory[43] (same line used for gene drive experiments). For these experiments, we waited until the injected G0 embryos became adults, then single-pair crossed them to white-eye mutant individuals. G1 progeny were screened for GFP expression, indicating transgene integration. By dividing the number of G0 crosses that gave rise to GFP-expressing progeny by the total number of G0 crosses, this strategy allowed us to evaluate single-germline transgenesis efficiency. The results are summarized in Supplementary Data 4.

**Gene drive experiments**. For gene drive experiments (Fig. 5), virgin F1 females carrying both the Cas9 construct and gRNA elements were single-pair crossed to *w*−/*w*− mutant males on the day of eclosion from the pulpal case. After 5 days, the F1 cross individuals were discarded, and the resulting F2 progeny was scored for

the presence of the GFP marker as an indicator of successful allelic conversion (Supplementary Data 5). Gene drive experiments (Fig. 5 and Supplementary Data 5) were performed in an ACL2 facility built for gene drive purposes at the Biological Sciences Department, University of California San Diego. In this facility, all experimental flies are frozen for 48 h before being removed from the room, autoclaved, and discarded as biohazardous waste.

**Graph generation and statistical analysis**. We used GraphPad Prism 7 to generate all our graphs. For statistical analysis, we used GraphPad Prism 7 and the Statkey analysis tool, version 2.1.1 [http://www.lock5stat.com/StatKey/index.html]. For Fig. 1b and Supplementary Fig. 3, we used a one-way ANOVA followed by Tukey's multiple comparison test to compare the allele editing efficiency (%) of $kh3$ target locus in Hsu cells (Supplementary Data 1). In Figs. 2 and 4, we performed a randomization test for a difference in proportions to evaluate differences in the proportion of independent germlines giving rise to transformants (positive for the respective fluorescent marker) in our two transgenesis conditions. For these analyses, we have performed 5000 and 20,000 randomizations of our data, respectively (Supplementary Data 2 and 4). We also performed one-way ANOVA and post hoc Dunnett's test for multiple comparisons to analyze our results in Fig. 5. The "Loop", "Mutation", and "Loop + Mutation" conditions were compared to the "Original" arrangement for their inheritance, cutting and allelic conversion rates separately (Supplementary Data 5).

**Reporting summary**. Further information on research design is available in the Nature Research Reporting Summary linked to this article.

## Data availability

The plasmid sequences of the constructs generated in this manuscript are either deposited into the GenBank database or available from the authors upon request. GenBank accession numbers for the deposited plasmids are the following: pVMG0173_Cq-Actin5c-Cas9 (MW925696), pVMG0193_Cq-Rpl40-Cas9 (MW925697), pVMG0213_Cq-vasa-Cas9 (MW925698), pVMG0212_Cq-nanos-cas9 (MW925699), pVMG0146_Cq-U6-1_2xBbsI-gRNA (MW925700), pVMG0217_Cq-U6-2b_2xBbsI-gRNA (MW925701), pVMG0218_Cq-U6-4_2xBbsI-gRNA (MW925702), pVMG0149_Cq-U6-6_2xBbsI-gRNA (MW925703), pVMG0164_Cq-U6-7_2xBbsI-gRNA (MW925704), pVMG0252_Cq-vasa-Cas9_cdHAs_O (MW925705), pVMG0109_CC-U6-3-w5_GFP_wHAs_O (MW925706), pVMG0302_Cq-U6-1_2xBbsI_gRNA-Loop (MW925707), pVMG0303_Cq-U6-6_2xBbsI_gRNA-Loop (MW925708), and pVMG0147_Cq-U6-2_2xBbsI-gRNA (MW925709). Selected plasmids have been deposited to the Addgene redistribution service, and are available for order by the community on the Addgene website (http://www.addgene.org/). Addgene identification numbers for the plasmids are as follows: pVMG0146_Cq-U6-1_2xBbsI-gRNA (169238), pVMG0149_Cq-U6-6_2xBbsI-gRNA (169323), pVMG0217_Cq-U6-2b_2xBbsI-gRNA (169339/), pVMG0173_Cq-Actin5c-Cas9 (169345), pVMG0193_Cq-Rpl40-Cas9 (169346), pVMG0213_Cq-vasa-Cas9 (169347), pVMG0212_Cq-nanos-cas9 (169348), pVMG0302_Cq-U6-1_2xBbsI_gRNA-Loop (169369), pVMG0303_Cq-U6-6_2xBbsI_gRNA-Loop (169370). Genbank and Addgene identification numbers are also available in Supplementary Fig. 1. All source data are provided with this paper; they cover the raw phenotypical scoring data collected, which is reported in the Supplementary Data 2–5 files in Microsoft Excel format (.xlsx). All other data are available upon request from the authors.

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

## Acknowledgements

We thank Anthony James, Vanessa Bottino Rojas, Kaycie Butler, and Ryan Staudt for comments and edits on the manuscript. We thank Anton Cornell and Nannan Liu for kindly providing *Culex quinquefasciatus* strains for this study. We thank Nelson Lau for kindly providing the Hsu cell line. We thank Emily Okamoto for helping with molecular experiments. The research reported in this manuscript was supported by the University of California, San Diego, Department of Biological Sciences, by the Office of the Director of the National Institutes of Health under award number DP5OD023098 to V.M.G., by a gift from the Tata Trusts of India to TIGS-UCSD, and by the NIH NIGMS grant P41GM132087 to N.P. N.P. is an investigator of HHMI.

## Author contributions

X.F. and V.M.G. conceived the project. X.F., V.L.D.A., E.M., A.L.B., N.P., and V.M.G. contributed to the design of the experiments. X.F., V.L.D.A., E.M., M.L., A.L.B., and V.M.G. performed the experiments and contributed to the collection and analysis of data. X.F., V.L.D.A., E.M., and V.M.G. wrote the manuscript. All authors edited the manuscript.

## Competing interests

V.M.G. is a founder of and has equity interests in Synbal, Inc. and Agragene, Inc., companies that may potentially benefit from the research results described in this manuscript. V.M.G. also serves on both the company's Scientific Advisory Board and the Board of Directors of Synbal, Inc. The terms of this arrangement have been reviewed and approved by the University of California, San Diego in accordance with its conflict of interest policies. X.F., V.L.D.A., E.M., M.L., A.L.B., and N.P. declare no competing interests.
