## [Peer Review File · Nature Communications]

Reviewers' Comments:

Reviewer #1:

Remarks to the Author:

Summary:

This manuscript by Feng et al. presents a first step towards gene drive in *Culex*. The authors test various genome editing reagents in cells, embryos and in a transgenic *Culex* strain that they generated via CRISPR-mediated homology-directed repair. The line expresses Cas9 under the *vasa* promoter and could be a valuable tool for other researchers, since it allows to achieve editing when injected with in-vitro transcribed gRNAs, as the authors demonstrate, or it could be crossed to gRNA-expressing lines in the future. The authors also test gRNA scaffold variants in the fruit fly.

The paper and figures are very nicely presented. However, I do have some comments.

First, a large part of the data is purely qualitative, and the authors could not and consequently did not provide any statistical analysis that would show significant effects. It is clear that for these messy types of experiments (e.g. mosaic genome editing events in injected embryos) much larger datasets are needed to obtain statistically relevant data. However, the authors do make statements that appear to indicate that they concluded more from these experiments than is warranted, given the nature of the data. For example, they state that "Our results suggest that the two gRNA scaffold variants tested can lead to a 4–5-fold increase in cutting activity". This is based on a comparison of 1/105 versus 3/55 recorded events which a simple Fisher's exact test will show to be not statistically significant difference. Another example is "These results show that the "Loop" gRNA scaffold variant can improve transgenesis in a second organism". Again, a quick test will reveal that there is no statistically significant difference here.

Regarding the scaffold variants. Some systems have achieved near-complete homing rates (e.g. *Anopheles gambiae*) with the canonical gRNA scaffold, so I'm missing the justification for why optimizing the gRNAs is a step to take in *Culex*. This is a minor point possibly, but in any case, the *Drosophila* part, which is supposed to analyse the effect of the gRNA scaffolds on drive efficiency (the statistical analysis is again missing), sits very uneasily next to the *Culex* part and it is not clear that there really is a connection is here.

I do not want to belittle the authors' achievements, after all *Culex* is technically more challenging to modify compared to other mosquito species, at least this is what the literature suggests. The authors without a doubt worked hard and can now provide a first set of useful tools for the endogenous use of CRISPR. It should be noted however that transgenic *Culex* mosquitoes have been generated previously via transposase-based methods and gene editing was achieved via REMOT edits (this needs to be cited) so it's not a completely uncharted territory.

Nature Communications seeks to publish important advances of significance to specialists within each field. I believe that if the authors had shown gene drive and determined the rate of homologous repair in-vivo in the *Culex* germline then the above would perhaps apply to this manuscript. Most steps towards this goal have been taken so it is somewhat strange that the authors didn't go this final step further. As it stands, and given the qualitative nature of the data, I do not feel that the paper is strong enough to warrant publication in Nature Communications.

Minor comments:

Line 57: maybe also cite the paper about ReMOT in *Culex* (<https://doi.org/10.1093/jme/tjab016>)

Line 64: generally defined as homology-directed repair (HDR) without "DNA"

Line 104: "While we could have synthesized the missing regulatory sequences" Odd sentence, it is not relevant what could have been done.

Line 108: the term "CRISPR transgene" is a bit confusing in this context, better to simply write "promoter"

Line 117/118: requires double-checking of the grammar in this sentence, in particular "and using"

Line 119: "A constant plasmid copy number of each gRNA promoter was transfected with an equal amount of Cas9-expressing plasmid"

This is hard to understand and can possibly be misunderstood. The explanation should be improved and expanded.

Line 127: "a two-fold increase in editing efficiency when extending the cell culture incubation time, confirming a positive correlation between the number of cell doublings and editing efficiency"

This doesn't seem to follow automatically. Even in non-dividing cells one would expect increased activity with increased time.

Line 147: The choice of the cardinal locus needs to be justified and explained for readers that aren't familiar with that system.

Fig 1e: Since U6:2b performed better than U6:2 in the cell line experiment, why was U6:2 chosen instead of U6:2b for further experiments?

Line 219: Why was it necessary to target this specific region of cardinal that showed polymorphisms? Wouldn't it be easier to target a more conserved region within the cardinal CDS and avoid the extra-work of making an isogenic line for this mutation?

Figure 2 b and Supplementary Figure 6 a: they are almost the same, better remove redundancies. Please include the sequences of the scaffold variants in FASTA format in the Supplement.

Sidenote to Table 1: It would be informative to provide data on transient marker expression of G0 survivors. Is this possible with Culex?

Sidenote to Line 273: Why were only 2 out of 4 G1 males taken to establish the line? It is usually difficult to get offspring from low numbers of parents.

Figure 3e: the coding is not consistent: better stick to "cd-, Cas9" also for the G0 (could colour the cd half light brown, similar to figure 2).

Line 327: Why were only the "original" and "loop" scaffolds evaluated in Drosophila, but not the "Loop + mutation", which seems better according to table 1?

Line 388: "-ly" missing in "significant"

Figure 5e: the blue triangle points upwards, whereas in the legend it points downwards

Line 471: Why/How would the scaffold variants mitigate the generation of resistant alleles?

Line 633/660/667/669/670/694/733: °C is missing in my version

Line 760: in "a" previous study

Line 773: mention the provider used for the generation of *Drosophila* injections

Supplementary Fig 1a:

I assume that what is drawn in light brown and marked as "gRNA" is in fact the "scaffold", since the gRNA will only be inserted afterwards via the 2 BbsI sites.

Supplementary Fig 6a: mostly redundant with Figure 2b.

The grey shaded area presumably represents the loops, but maybe better to indicate it in the legend.

Supplementary Data to Figure 3e:

The section about the 2nd BF does not contain any data. Better to remove this section entirely.

Supplementary Data Crispresso Batch in vivo:

There is "(?)" written instead of the actual percentage.

Reviewer #2:

Remarks to the Author:

In the manuscript, "Optimized CRISPR tools and site-directed transgenesis in *Culex quinquefasciatus* mosquitos for gene drive development", the authors present multiple novel implementations of CRISPR/Cas9 expression in this organism. The authors show that Cas9 can be expressed when genomically integrated and can be expressed in a heritable fashion. Additionally, they show that this germline Cas9 can be used to generate CRISPR targeted genomic cleavage. The authors stop short in demonstrating that these Cas9 integrated *Culex* strains are able to drive genes through success generations, however they show that CRISPR tools trialed in *Culex* improve driving ability in *drosophila*. This work progresses CRISPR tools in *Culex* and gene drive tools in *drosophila* and provides additional strains for the exploration of gene drive work in *Culex* species.

Major concerns:

My main concern about this work is the way the Cas9/kh3 disrupted cross work is presented as described in figure 3e. As described, it appears that heterozygous or homozygous Cas9 expressing eggs were injected with either plasmid or in vitro transcribed gRNA targeting the kh3 gene and then these strains were crossed with kh- homozygous strains to interrogate the frequency of cutting that occurred after injection in the G0 individuals. To clarify this experiment, I would explicitly state that these occurrences were not the result of gene driven cleavage of kh in the G1, but instead a result of modification of the kh by germline expressed Cas9 in the G0. This is described; however, it warrants further clarification to not obfuscate that this modification was not due to true gene drive activity.

As an additional point, an additional experiment to show the ability of this integrated Cas9 to drive a gRNA knockout or gene to a successive generation would provide a capstone to this work and strengthen the utility and impact greatly. I would greatly encourage this experiment or describe why this experiment was not completed.

Lastly, it should be made clear in that the further optimization of CRISPR tools for gene drive development were only fully tested in a gene drive format within *drosophila*, perhaps by modifying the title.

Minor concerns:

Please clarify the number of larvae examined for deep sequencing percentages in in figure 1 F.

Authors' response to the Reviewers

We thank both reviewers for their feedback on our manuscript, and for recognizing the value of the reagents generated in this work for the *Culex* research community. The Reviewers' assessment helped us to improve the manuscript substantially, and we hope to have addressed all the concerns raised and answered each point to satisfaction. We are providing a revised version of the manuscript, including the suggested edits and revisions, with comments highlighting the edits addressing each specific numbered comment. Please find here below point-by-point responses in blue text.

Reviewer #1 (Remarks to the Author):

Summary:

This manuscript by Feng et al. presents a first step towards gene drive in *Culex*. The authors test various genome editing reagents in cells, embryos and in a transgenic *Culex* strain that they generated via CRISPR-mediated homology-directed repair. The line expresses Cas9 under the *vasa* promoter and could be a valuable tool for other researchers, since it allows to achieve editing when injected with in-vitro transcribed gRNAs, as the authors demonstrate, or it could be crossed to gRNA-expressing lines in the future. The authors also test gRNA scaffold variants in the fruit fly.

The paper and figures are very nicely presented. However, I do have some comments.

1. We thank the reviewer for appreciating our effort on presenting our findings in the manuscript, and underscoring the value of the reagents generated for the research community.

First, a large part of the data is purely qualitative, and the authors could not and consequently did not provide any statistical analysis that would show significant effects. It is clear that for these messy types of experiments (e.g. mosaic genome editing events in injected embryos) much larger datasets are needed to obtain statistically relevant data. However, the authors do make statements that appear to indicate that they concluded more from these experiments than is warranted, given the nature of the data. For example, they state that "Our results suggest that the two gRNA scaffold variants tested can lead to a 4–5-fold increase in cutting activity". This is based on a comparison of 1/105 versus 3/55 recorded events which a simple Fishers exact test will show to be not statistically significant difference.

2. Regarding the qualitative nature of some of our data. The Reviewer is correct in stating that some of our data is qualitative, such as the evaluation of gRNA expression in injected embryos. After we generated the gene editing data in cells, which was performed in triplicates and for which we are reporting p-values for the relevant comparisons, we proceeded to confirm our observations using transient expression in embryos. This analysis aimed at obtaining a yes/no answer on whether the generated constructs were also able to express gRNAs in embryos as they did in cells. Because this experiment was only confirmatory, we generated only one replicate in the embryo experiment, as it would have required the injection of hundreds of additional eggs for each data point. Also, since we observed significant differences in the cell analysis and the observed embryo data was correlating well with it, we decided that this information was enough to proceed to the HDR transgenesis step.

Regarding the statement "Our results suggest that the two gRNA scaffold variants tested can lead to a 4–5-fold increase in cutting activity," referring to the data in **Fig. 2d** and **Table 1**. We agree that perhaps this statement is too strong given that the "4-5-fold" difference is based on a

limited number of G0 germlines analyzed and a P-value = 0.055. Also, while the reviewer is correct in stating that “based on a comparison of 1/105 versus 3/55 recorded events which a simple Fishers exact test will show to be not statistically significant difference”, when we instead compare the overall number of G1 edited or transgenic animals obtained, we do observe a significant difference. We have updated **Table 1** in the revised manuscript and we also modified the text accordingly to explain these observations. For the reviewer’s reference here is a copy of **Table 1**:

HR Plasmid	Injected G0			Efficiency (per G0 germline)*		Overall efficiency (out of total G1s)#	
	Injected eggs	Adult survivors	Survival	Cutting: cd-/cd- (%)	Transgenesis DsRed+ (%)	Cutting: cd-/cd- (%)	Transgenesis DsRed+ (%)
“Original”	815	105	12.88%	1/105 (0.95%)	0/105 (0%)	6/5377 (0.11%)	0/5377 (0%)
“Loop”	1480	55	3.72%	3/55 (5.45%) (p-value=0.055) ns	1/55 (1.82%) (p-value=0.167) ns	25/3148 (0.79%) (p-value<0.0001) ****	4/3148 (0.13%) (p-value=0.014) *
“Loop+Mutation”	1151	24	2.09%	1/24 (4.17%) (p-value=0.188) ns	1/24 (4.17%) (p-value=0.025) *	22/1126 (1.95%) (p-value<0.0001) ****	4/1126 (0.36%) (p-value=0.0008) ***

Additionally, while we do observe a trend in the G0 germline data and we have observed a significant increase for the G1 data, we still believe that using the number of positive G0 germlines remains a more accurate value for comparisons. This is because the “Overall efficiency” measured in the G1 could carry a bias due to the assortative mating in each pool and the number of eggs laid per female. Since we did not want to overstate our findings, we have avoided making strong statements regarding the G1 Overall efficiency, and focused on comparing the positive G0 germlines between conditions which clearly display a consistent trend in line with 1) the *Drosophila* transgenesis (**Fig.4**), 2) the *Drosophila* gene drive data (**Fig.5**) and 3) previous reports (Dang et al. 2015).

Based on the reviewer’s comments and the reasoning above we have modified the text as follows:

ORIGINAL:

“Our results suggest that the two gRNA scaffold variants tested can lead to a 4–5-fold increase in cutting activity, which in turn should promote the likelihood of obtaining HDR events in transgenesis efforts.”

NEW VERSION:

“Additionally, when we compare the overall number of cut or transgenic G1 animals obtained using the modified scaffolds to the “Original” scaffold, we observe a significant increase in both editing and transgenesis (P-values in Table 1). These combined results support the hypothesis that the two gRNA scaffold variants tested can lead to a substantial increase in cutting activity, in line with previous reports⁴³. In turn, a higher cutting rate should promote an increased likelihood of obtaining HDR events in transgenesis efforts.”

Another example is “These results show that the “Loop” gRNA scaffold variant can improve transgenesis in a second organism”. Again, a quick test will reveal that there is no statistically significant difference here.

- Regarding the statement “These results show that the “Loop” gRNA scaffold variant can improve transgenesis in a second organism” which refers to **Fig. 4** where we evaluated the effect of the “loop” modification in *Drosophila* transgenesis experiments. Our original manuscript version already included a Randomization Test for a Difference in Proportions to evaluate these differences (**Supplementary Data 4**). In this analysis, our null hypothesis is: “There are no differences in transgenesis between the “original” and “Loop” constructs”. Instead, our alternative hypothesis is: “The “loop” construct leads to an increased transgenesis rate” which led us to generate a 1-tail test. Given that we recovered a P-value = 0.046, we reject the null hypothesis and conclude that there is a significant increase compared to the “Original” gRNA. After the reviewer’s comment we have generated a second Randomization test, increasing the randomizations number from 5,000 to 20,000, and obtained a P-value = 0.039, which we have modified in the text as it is lower than we previously obtained and better supports our claims as it is more robust (20,000 Randomizations).

To mirror the data presentation of the *Culex* transgenesis experiments we have generated an additional **Table 2** summarizing the results of the *Drosophila* transgenesis experiment. The new table also includes the P-values for each comparison as done for **Table 1**. For the reviewer’s reference here is a copy of **Table 2**:

HDR Plasmid	Injected G0 individuals	Efficiency (per G0 germline)*	Overall efficiency (out of total G1s)#
	G0 Single-pair crosses	Vials generating GFP+ (%)	Overall G1 GFP+ recovered (%)
“Original”	41	11/41 (26.19%)	185/1982 (9.5%)
“Loop”	48	21/48 (44%) (p-value=0.039) *	279/2379 (13.5%) (p-value=0.0053) **

Based on the reviewer’s comment, it seems that perhaps the reviewer used a different statistical analysis and obtained a slightly different P-value for this comparison. Therefore we have evaluated each statistical comparison from the *Culex* and *Drosophila* transgenesis experiments with 1) a Test for a Difference in Proportions, which we have been using for our statistical analysis; 2) Fisher’s Exact Test; and 3) Chi². Since we have an expectation about the direction of the difference, based on previous data (Dang et al. 2015), these tests are a 1-tail analysis. For the Reviewer’s reference, and for full transparency we include this analysis below. Additionally, in case of publication it will be available to the readers who will have access to the different statistics used to compare the data. Green indicates P-values supporting significant differences, yellow indicates P-values close to the arbitrary 0.05 cut-off (see next page).

						P-values (1-tail Tests)		
Culex	Cutting efficiencies		cd- mutants	Individuals non-edited (wt)	Total	Randomization	Fisher	Chi^2
	Cutting germline (per G0 germline) - Table 1, Fig.2D	Original	1	104	105	-	-	-
		Loop	3	52	55	0.055	0.1177	0.0416*
		L+M	1	23	24	0.188	0.3387	0.1251
	Overall cutting (G1 scored individuals) - Table 1	Original	6	5371	5377	-	-	-
		Loop	25	3123	3148	<0.0001****	<0.0001****	<0.0001****
		L+M	22	1104	1126	<0.0001****	<0.0001****	<0.0001****
	HDR efficiency		DsRed+	DsRed -	Total	Randomization	Fisher	chi2
	Transgenesis efficiency (per G0 germline) - Table 1	Original	0	105	105	-	-	-
		Loop	1	54	55	0.167	0.3437	0.0829
		L+M	1	23	24	0.025*	0.186	0.0179*
	Overall transgenesis (G1 scored individuals) - Table 1	Original	0	5377	5377	-	-	-
Loop		4	3144	3148	0.014*	0.0186*	0.0045**	
L+M		4	1122	1126	0.0008***	0.0009***	<0.0001****	
Drosophila	HDR efficiency		GFP+	GFP-	Total	Randomization	Fisher	chi2
	Transgenesis (per G0 germline) - Table 2, Fig. 4C	Original	11	31	42	-	-	-
		Loop	21	27	48	0.039*	0.0643	0.0413*
	Overall Transgenesis (G1 scored individuals) - Table 2	Original	185	1797	1982	-	-	-
		Loop	279	2100	2379	0.0053**	0.0060**	0.0053**

Regarding the scaffold variants. Some systems have achieved near-complete homing rates (e.g. *Anopheles gambiae*) with the canonical gRNA scaffold, so I'm missing the justification for why optimizing the gRNAs is a step to take in *Culex*. This is a minor point possibly, but in any case, the *Drosophila* part, which is supposed to analyse the effect of the gRNA scaffolds on drive efficiency (the statistical analysis is again missing), sits very uneasily next to the *Culex* part and it is not clear that there really is a connection is here.

- Regarding the justification for testing the optimized gRNAs in *Culex*. Before our work there were no validated resources to express CRISPR components in *Culex*. And while the reviewer is correct in stating that other “systems have achieved near-complete homing rates (e.g. *Anopheles gambiae*) with the canonical gRNA scaffold”, in other systems such as *Aedes aegypti* a more modest gene drive inheritance was observed (~70%, Li., et al 2020 *eLife*). Because our ultimate goal of building a gene drive requires HDR transgenesis, we wanted to maximize our chances of success by lowering the variables in our system and using optimized components. We did so by: 1) generating a line that is homozygous at the *cardinal* locus for a given allele so that the homology arms of our construct would perfectly match each copy of the genome; 2) generating a single Cas9-expressing, gRNA-expressing, HDR-template plasmid so that each germline cell receiving a copy of the injected plasmid would have all the necessary components needed for transgenesis, and 3) including the scaffold variants in our analysis, which in preliminary studies in the lab showed higher activity compared to the “Original” gRNA scaffold. Indeed, we believe that this last addition is the principal reason that we were successful in obtaining transgenics. We want to highlight that in each injection using the “Original” gRNA we could not recover any transgenic animals and we observed cutting in only one germline (out of 105+36=141 germlines tested). In contrast we were able to recover HDR-transgenics in four independent instances when using the modified scaffold variants (**Supplementary Data 1**).

Regarding the statistics for the gene drive experiments. As we mentioned in the response to the previous comments, the statistical analysis regarding the *Drosophila* gene drive experiments was already present in **Supplementary Data 5**, and it is referenced in the text. Additionally, following the Reviewer’s comment, we have included the P-values for these comparisons in **Fig. 5d**, to better highlight the differences observed.

Regarding the connection between the *Culex* and *Drosophila* studies, we decided to combine these experiments for two main reasons:

A) We decided to use *Drosophila*, for which single pair genetic crosses are more amenable than mosquitoes, to generate a deeper evaluation of the effect of the “loop” variant on transgenesis. We have modified the text to make this justification more obvious. An increase in transgenesis in a second organism would further support the trend observed for the *Culex* transgenesis experiments when using the scaffold variants. We believe that combining *Culex* and *Drosophila* data also supports the generalization of the effect of the “loop” gRNA scaffold variant’s potential to boost transgenesis in other organisms. Research groups working with pioneer organisms could benefit from an increased activity of these gRNA variants in both editing and transgenesis efforts.

B) The gene drive portion was added because our ultimate goal is to use the generated tools to build gene drives in *Culex*, and we wanted to make sure that these modified scaffolds would lead to an improvement in gene drive and would not be detrimental instead. Since we have become proficient in the generation of split gene drive systems, we have employed it as a tool for this sort of evaluation. In addition, the very sensitive gene drive system allowed us to generate data further supporting the findings described for the *Culex* and *Drosophila* transgenesis efforts.

I do not want to belittle the authors achievements, after all *Culex* is technically more challenging to modify compared to other mosquito species, at least this is what the literature suggests. The authors without a doubt worked hard and can now provide a first set of useful tools for the endogenous use of CRISPR. It should be noted however that transgenic *Culex* mosquitoes have been generated previously via transposase-based methods and gene editing was achieved via REMOT edits (this needs to be cited) so it’s not a completely uncharted territory.

5. We thank the reviewer for recognizing how challenging working with *Culex* mosquitoes is compared to other species. The reviewer is correct in stating that *Culex quinquefasciatus* transgenesis has been achieved before using transposons, indeed we cited two manuscripts in the introduction which, to our knowledge, are the only two examples of transgenesis in this species prior to this work. Three other groups beyond ours have shown CRISPR gene editing in *Culex quinquefasciatus* using Cas9 protein and *in-vitro* transcribed gRNAs; all these manuscripts are also cited in the introduction. Differently from these manuscripts, in this work we are genetically encoding the gRNA and Cas9 to generate genome edits, and later use these tools to demonstrate the feasibility of HDR-based site-directed transgenesis. Both aspects have not yet been published in peer-reviewed journals. Luke Alphey Lab has also achieved HDR-based transgenesis in *Cx. quinquefasciatus*, but since this work has not yet been peer-reviewed, we decided to cite it in the discussion rather than the introduction. We thank the reviewer for pointing out this recent manuscript showing REMOT working in the sister species *Culex pipiens pallens*; we were not aware of this work at the time, as it was published after our submission to *Nature Communications*. We have now added this reference in the introduction.

Nature Communications seeks to publish important advances of significance to specialists within each field. I believe that if the authors had shown gene drive and determined the rate of homologous repair in-vivo in the *Culex* germline then the above would perhaps apply to this manuscript. Most steps towards this goal have been taken so it is somewhat strange that the authors didn’t go this final step further. As it stands, and given

the qualitative nature of the data, I do not feel that the paper is strong enough to warrant publication in Nature Communications.

6. While our group is working towards gene drive applications in *Culex quinquefasciatus*, the first step towards this achievement is i) the generation of validated transgenes for the expression of gRNA/Cas9 and ii) the development of HDR based transgenesis. The scope of this work was indeed to achieve these two goals. As Reviewer 1 points out above “*Culex is technically more challenging to modify compared to other mosquito species*”, which is what led us to evaluate scaffold variants for increasing editing efficiency. As also mentioned in previous comments, we believe our extended analysis in *Drosophila melanogaster* supports the application of these optimized gRNAs in other organisms for both editing and gene drive development. As Reviewer #1 states we “*without a doubt worked hard and can now provide a first set of useful tools for the endogenous use of CRISPR*” not only for the development of gene drive but for use to the broader community working with *Culex*.

This is why we believe that our study is an important advance of significance to the research community working with this challenging vector, and that our gRNA scaffold variant analysis could have broader implications beyond *Culex* research. In fact, we have already been contacted by several groups, and we shared our reagents with multiple labs. We are also preparing to ship our *vasa*-Cas9 mosquito line nationally and internationally, and consulted with groups working with *Culex*, *Anopheles* and ticks regarding the use of the tested gRNA variants in these systems. All these interactions happened after posting this work in biorXiv (~1.5 months ago), underscoring the value of this work to the broader community.

Indeed, we are confident the publication of this manuscript in *Nature Communications* will be of interest for the research community and will promote the development of CRISPR-based technologies in *Culex quinquefasciatus* mosquitoes.

Minor comments:

Line 57: maybe also cite the paper about ReMOT in *Culex* (<https://doi.org/10.1093/jme/tjab016>)

7. We thank the reviewer for this suggestion. We have added this reference in the introduction.

Line 64: generally defined as homology-directed repair (HDR) without “DNA”

8. Agreed. We have removed “DNA” from the sentence as suggested by the Reviewer.

Line 104: “While we could have synthesized the missing regulatory sequences” Odd sentence, it is not relevant what could have been done.

9. Agreed. The sentence has been modified in the manuscript.

Line 108: the term “CRISPR transgene” is a bit confusing in this context, better to simply write “promoter”

10. Agreed. “CRISPR transgene” was replaced by “promoter”.

Line 117/118: requires double-checking of the grammar in this sentence, in particular “and using”

11. Agreed. This sentence has been modified in the manuscript.

Line 119: “A constant plasmid copy number of each gRNA promoter was transfected with an equal amount of Cas9-expressing plasmid” This is hard to understand and can possibly be misunderstood. The explanation should be improved and expanded.

12. We thank the reviewer for pointing this out, we have modified the sentence in the text to clarify the meaning.

Line 127: “a two-fold increase in editing efficiency when extending the cell culture incubation time, confirming a positive correlation between the number of cell doublings and editing efficiency”

This doesn't seem to follow automatically. Even in non-dividing cells one would expect increased activity with increased time.

13. We thank the reviewer for this comment, we have clarified this in the text and modified it as follows:

ORIGINAL: “Interestingly, for the Cq-U6:1 promoter we observed a two-fold increase in editing efficiency when extending the cell culture incubation time, confirming a positive correlation between the number of cell doublings and editing efficiency ”

NEW VERSION: “Interestingly, for the Cq-U6:1 promoter we observed a two-fold increase in editing efficiency when extending the culturing time after transfection, confirming that a longer exposure to the CRISPR reagents leads to increased genome editing in cells.”

Line 147: The choice of the cardinal locus needs to be justified and explained for readers that aren't familiar with that system.

14. We have modified this portion of the text to include a justification for the choice of the *cardinal* locus for our analysis:

“We chose the cardinal gene for this analysis for three reasons: 1) we have previously validated editing at this locus using cd1-gRNA and built a cardinal- homozygous line ³¹, which we will use later in this work; 2) cardinal- homozygous mutants display a visible phenotype, a lighter-than-wild-type, red-eye, which darkens over time and potentially leads to an almost-wild-type fitness of an eventual homozygous transgenic line; and 3) we could directly use the reagents validated in embryos for the next step of this project, aiming to obtain HDR-based transgenesis.”

Fig 1e: Since U6:2b performed better than U6:2 in the cell line experiment, why was U6:2 chosen instead of U6:2b for further experiments?

15. Based on our cell data, we selected four U6s with variable editing efficiencies (U6:6 the highest, U6:1 intermediate, U6:2 low, and U6:4 extremely low activities) to evaluate whether the *in vivo* values displayed a trend that was correlating with the cell data. Indeed we do observe a general

trend in which in both experiments U6:1 and U6:6 have the highest activities, U6:2 has an intermediate one and U6:4 show the lowest editing.

Line 219: Why was it necessary to target this specific region of cardinal that showed polymorphisms? Wouldn't it be easier to target a more conserved region within the cardinal CDS and avoid the extra-work of making an isogenic line for this mutation?

16. Our laboratory *Culex* lines display high polymorphism rates at all seven genomic locations tested in our previous work (Feng. et al. 2021 BiorXiv). At first we decided to target this region of *cardinal* in a region of CDS that displayed low polymorphism, although it still displayed SNPs nearly every ~50bp, which could have impeded HDR efficiency in some of our preliminary attempts (data not reported in this manuscript). To further increase our chance to get HDR working, we decided to generate a *Culex* line that was isogenic at *cardinal* locus (See methods and **Supplementary Figure 4**).

Figure 2 b and Supplementary Figure 6 a: they are almost the same, better remove redundancies. Please include the sequences of the scaffold variants in FASTA format in the Supplement.

17. Regarding the redundancy between **Fig. 2b** and **Supplementary Fig. 6**. It is true that these two figures are somewhat redundant, and have some overlapping information. However, in **Supplementary Fig. 6a** we highlight specific sequences that we mention in the discussion section, in an attempt to explain our results in comparison with previously published work (**Supplementary Fig. 6b**). We decided to include **Supplementary Fig. 6** as part of the supplement because we believed that adding additional sequences and highlighting the sequences mentioned in the discussion within **supplementary Fig. 6** will better provide useful information to readers without potential confusion.

The FASTA sequences have been added in the Supplementary Methods section of the Supplementary Information File. Additionally, key plasmid sequences will be submitted to Genbank ahead of publication, and Genbank IDs will be added in the final version of this manuscript.

Sidenote to Table 1: It would be informative to provide data on transient marker expression of G0 survivors. Is this possible with *Culex*?

18. Yes, in most cases, injected G0 animals displayed fluorescence produced by the transient expression of the markers present on the plasmid. Although, instead of observing the whole body displaying fluorescence (eg. Opie2 promoter), the transient expression in G0 only shows fluorescence in some dots or patches distributed in variable regions. Because of this variability, we did not track the transient marker expression in the G0, and we would not be able to provide this information without repeating the experiments.

Sidenote to Line 273: Why were only 2 out of 4 G1 males taken to establish the line? It is usually difficult to get offspring from low numbers of parents.

19. We agree with the Reviewer that it is difficult to get offsprings from lower numbers of parents. Although, since the four G1 transgenic males we obtained might derive from different G0 within the pool, to ensure clean transgene isolation we actually single-mated each transgenic male to

multiple females. Out of these 4 crosses, only 2 gave offsprings, which we then used to establish two separate Cas9 isogenic lines. We have modified the text to clarify this point.

Figure 3e: the coding is not consistent: better stick to “cd-, Cas9” also for the G0 (could colour the cd half light brown, similar to figure 2).

20. We thank the reviewer for this suggestion, we have modified the figure to maintain a consistent iconography.

Line 327: Why were only the “original” and “loop” scaffolds evaluated in *Drosophila*, but not the “Loop + mutation”, which seems better according to table 1?

21. At first we generated and tested four transgenics with the different gRNA scaffolds to obtain the *Drosophila* gene drive data (**Fig. 5**), but did not keep a detailed record of the transgenesis efficiency for each condition. After seeing that the “Loop” variant displayed a significantly higher editing and HDR conversion efficiency in gene drive (**Fig. 5**), and the same “Loop” variant also improved *Culex* transgenesis (**Fig. 2**), we decided to test whether this was also true in *Drosophila* by repeating the fruit fly transgenesis only for the “Original” and “Loop” constructs (**Fig. 4**).

Line 388: “-ly” missing in “significant”

22. Thanks for catching this typo, it was corrected in the manuscript.

Figure 5e: the blue triangle points upwards, whereas in the legend it points downwards

23. The triangle was modified in the figure, as suggested by the Reviewer.

Line 471: Why/How would the scaffold variants mitigate the generation of resistant alleles?

24. A higher ratio of HDR-converted to NHEJ events such as shown for the “Loop” variant in **Fig. 5d** (82% Conversion rate vs. 74% for the Original) would lead to a lowered generation of resistant alleles as a gene drive spreads within a population and therefore, to a more efficient spread. The Reviewer was correct in pointing this out as the two other variants tested do not show an increased HDR-rate and therefore, our previous statement would have been slightly misleading. We have adjusted the text, clarifying that our claim would apply only to the “Loop” variant.

Line 633/660/667/669/670/694/733: °C is missing in my version

25. Thanks for catching this typos. Something went wrong on the character conversion during the upload to *Nature Communications*, and we did not catch the mistake in the proofs. We fixed this issue in all mentioned places, by using two separate characters.

Line 760: in “a” previous study

26. We fixed this typo.

Line 773: mention the provider used for the generation of *Drosophila* injections

27. We added the provider used for *Drosophila* injections as suggested.

Supplementary Fig 1a:

I assume that what is drawn in light brown and marked as “gRNA” is in fact the “scaffold”, since the gRNA will only be inserted afterwards via the 2 BbsI sites.

28. The text in the figure was edited from “gRNA” to “gRNA scaffold”

Supplementary Fig 6a: mostly redundant with Figure 2b.

The grey shaded area presumably represents the loops, but maybe better to indicate it in the legend.

29. The redundancy between **Fig. 2b** and **Supplementary Fig. 6** was already addressed in Comment #17.

We have added a sentence in both figure legends to indicate the function of the gray shaded area: *“The grey shaded area in the figure highlights the synthetic portions of the gRNA variants that were introduced to link the crRNA and the tracrRNA.”*

Supplementary Data to Figure 3e: The section about the 2nd BF does not contain any data. Better to remove this section entirely.

30. Agreed. This section has been deleted from **Supplementary Data 3**.

Supplementary Data Crispresso Batch in vivo:

There is “(?)” written instead of the actual percentage.

31. Thanks for catching this oversight, we have added percentages in the **Supplementary Data 1**.

Reviewer #2 (Remarks to the Author):

In the manuscript, "Optimized CRISPR tools and site-directed transgenesis in *Culex quinquefasciatus* mosquitos for gene drive development", the authors present multiple novel implementations of CRISPR/Cas9 expression in this organism. The authors show that Cas9 can be expressed when genomically integrated and can be expressed in a heritable fashion. Additionally, they show that this germline Cas9 can be used to generate CRISPR targeted genomic cleavage. The authors stop short in demonstrating that these Cas9 integrated *Culex* strains are able to drive genes through success generations, however they show that CRISPR tools trialed in *Culex* improve driving ability in *Drosophila*. This work progresses CRISPR tools in *Culex* and gene drive tools in *Drosophila* and provides additional strains for the exploration of gene drive work in *Culex* species.

1. We thank the reviewer for appreciating the value of our work, and our efforts to generate useful tools for CRISPR editing *Culex* species.

Major concerns:

My main concern about this work is the way the Cas9/kh3 disrupted cross work is presented as described in figure 3e. As described, it appears that heterozygous or homozygous Cas9 expressing eggs were injected with either plasmid or in vitro transcribed gRNA targeting the kh3 gene and then these strains were crossed with kh-homozygous strains to interrogate the frequency of cutting that occurred after injection in the G0 individuals. To clarify this experiment, I would explicitly state that these occurrences were not the result of gene driven cleavage of kh in the G1, but instead a result of modification of the kh by germline expressed Cas9 in the G0. This is described; however, it warrants further clarification to not obfuscate that this modification was not due to true gene drive activity.

2. We thank the Reviewer for this comment, and we acknowledge that **Fig. 3** and its results section needed further clarification on this point. We modified the figure to indicate that the germline editing happens in the G0 injected individuals. We also modified the nomenclature of the *kh* alleles to make it clear that one allele comes from the edited G0 germline while the other comes from the mutant *kh-/kh-* parent. These modifications should clarify to the reader that the phenotypes observed in the G1 are a reflection of the G0 germline editing and not gene drive. We have also modified the main text accordingly to better explain this point (underscored in the text below):

"We injected eggs from a heterozygous Cas9 line with an in-vitro transcribed (IVT) or a plasmid-expressed kh3-gRNA previously validated within our lab³¹. Since the eggs were obtained from a heterozygous line, we screened the surviving adults for the presence of the DsRed marker and discarded any animals without fluorescence prior to crosses with a homozygous kh-/kh- mutant line (Fig. 3e)³¹. This cross was performed to evaluate the occurrence of genome editing events happening in the germline of G0 injected animals. Successful edits in the G0 germline at the kh locus (kh, Fig. 3e), would combine with kh- alleles provided by the homozygous kh-/kh- line, and lead to white-eye phenotype in the G1 (kh*/kh-, Fig. 3e). We then screened the resulting G1 offspring for presence of the kh- white-eye phenotype, which would indicate a successful Cas9/kh3-gRNA-driven kh+ allele disruption in the G0 germline. We recovered G1 white-eye animals in both conditions tested, and molecularly validated the editing in these animals by sequencing PCR amplicons (Supplementary Fig. 5). While the IVT-gRNA yielded ~50% editing efficiency, the injection of the plasmid gRNA resulted in the recovery of ~1% of mutant G1s (Fig. 3e, Supplementary Data 3). It is possible that Cas9 protein preloaded in the egg, can readily bind the injected IVT-gRNA leading to early and*

efficient editing, while the plasmid could result in a more gradual production of the gRNA leading to lower editing in the germline. While these two strategies are not directly comparable due to the differences in gRNA delivery, both results confirm that the vasa promoter in our transgene is capable of driving efficient expression of Cas9 in the germline and producing genome edits at the kh locus.

As an additional point, an additional experiment to show the ability of this integrated Cas9 to drive a gRNA knockout or gene to a successive generation would provide a capstone to this work and strengthen the utility and impact greatly. I would greatly encourage this experiment or describe why this experiment was not completed.

3. Indeed, the experiment proposed by the reviewer would demonstrate the feasibility of gene drive in *Culex*. It is an experiment that we are very interested in performing and we will be working towards in the future. This manuscript's scope was instead to lay the foundation for future gene drive work in *Culex quinquefasciatus*, and this is why we focused on achieving two main goals: i) the generation of validated transgenes for the expression of gRNA/Cas9; and ii) the development of HDR based transgenesis. We believe that our work as it stands provides the research community with validated toolkit and strategies for the genome editing of *Culex* mosquitoes using CRISPR. In fact, as mentioned earlier in response to Reviewer #1 Comment#6, we have been contacted by several groups and already shared our reagents with multiple labs. We are also preparing to ship our *vasa*-Cas9 mosquito line nationally and internationally, and consulted with groups working with *Culex*, *Anopheles* and ticks regarding the gRNA variants. All these interactions happened after posting this work in bioRxiv (~ 1.5 months ago), underscoring the value of this work for the community beyond the development of gene drive.

Lastly, it should be made clear in that the further optimization of CRISPR tools for gene drive development were only fully tested in a gene drive format within *Drosophila*, perhaps by modifying the title.

4. While we used *Drosophila* as a tool to evaluate aspects of our research that arose during experimentation, with our title, we were hoping to convey to the reader both the value of the reagents described in this manuscript for *Culex quinquefasciatus* research, as well as capturing their potential for future gene drive development. As previously phrased, the title could have been misinterpreted by the reader, and upon the Reviewer's comment we have modified it to better reflect our intention:

ORIGINAL:

"Optimized CRISPR tools and site-directed transgenesis in Culex quinquefasciatus mosquitoes for gene drive development."

NEW TITLE:

"Optimized CRISPR tools and site-directed transgenesis towards gene drive development in Culex quinquefasciatus mosquitoes."

Minor concerns:

Please clarify the number of larvae examined for deep sequencing percentages in in figure 1 F.

5. We have added the number of larvae (30) that we pooled for the deep sequencing analysis in **Fig. 1**. The information is also available in the Methods section.

Reviewer #1 (Remarks to the Author):

The authors did a good job addressing the minor comments.
Overall, my view on the paper hasn't changed.

I would be happy to re-review the paper if some data on transmission/homing is provided.

Reviewer #2 (Remarks to the Author):

I think the authors have improved the manuscript and transparency of experiments. Their work adds to the CRISPR/Cas gene drive toolkit in a difficult to engineer species.

AUTHORS' RESPONSE TO THE REVIEWERS' COMMENTS

Reviewer #1 (Remarks to the Author):

The authors did a good job addressing the minor comments.
Overall, my view on the paper hasn't changed.

I would be happy to re-review the paper if some data on transmission/homing is provided.

Reviewer #2 (Remarks to the Author):

I think the authors have improved the manuscript and transparency of experiments. Their work adds to the CRISPR/Cas gene drive toolkit in a difficult to engineer species.

We thank both reviewers for their feedback which has helped us to greatly improve the manuscript. It seems that both reviewers have reviewed the updated manuscript and are satisfied with our answers to their previous concerns.